# Gene expression variability in human and chimpanzee populations share common determinants

Benjamin Jung Fair[1]*, Lauren E Blake[2], Abhishek Sarkar[2], Bryan J Pavlovic[3], Claudia Cuevas[2], Yoav Gilad[1,2]*

[1]Department of Medicine, University of Chicago, Chicago, United States;
[2]Department of Human Genetics, University of Chicago, Chicago, United States;
[3]Department of Neurology, University of California, San Francisco (UCSF), San Francisco, United States

**Abstract** Inter-individual variation in gene expression has been shown to be heritable and is often associated with differences in disease susceptibility between individuals. Many studies focused on mapping associations between genetic and gene regulatory variation, yet much less attention has been paid to the evolutionary processes that shape the observed differences in gene regulation between individuals in humans or any other primate. To begin addressing this gap, we performed a comparative analysis of gene expression variability and expression quantitative trait loci (eQTLs) in humans and chimpanzees, using gene expression data from primary heart samples. We found that expression variability in both species is often determined by non-genetic sources, such as cell-type heterogeneity. However, we also provide evidence that inter-individual variation in gene regulation can be genetically controlled, and that the degree of such variability is generally conserved in humans and chimpanzees. In particular, we found a significant overlap of orthologous genes associated with eQTLs in both species. We conclude that gene expression variability in humans and chimpanzees often evolves under similar evolutionary pressures.

**\*For correspondence:**
bjf79@uchicago.edu (BJF);
gilad@uchicago.edu (YG)

**Competing interests:** The authors declare that no competing interests exist.

## Introduction

Variation in gene expression underlies many phenotypic differences between and within species. Gene expression itself is a quantitative phenotype that is subject to both random drift and natural selection. A deeper understanding of how natural selection shapes gene expression across primates is central to our understanding of human evolution, and may also elucidate the mechanistic basis for variation in quantitative traits and disease risk within species.

Several studies have used a comparative transcriptomics approach across species and tissues to identify genes whose expression patterns are consistent with the action of natural selection (*Barbosa-Morais et al., 2012*; *Brawand et al., 2011*; *Merkin et al., 2012*). For example, a pattern of highly similar gene expression levels in all primates may be consistent with the action of stabilizing selection on gene regulation, and potentially, negative selection on the corresponding regulatory elements. Indeed, many studies have found that the expression of most genes evolves slower than expected under neutrality (*Chan et al., 2009*; *Khaitovich et al., 2006*; *Khan et al., 2013*; *Lemos et al., 2005*; *Merkin et al., 2012*; *Romero et al., 2012*). Conversely, genes that show a reduced or elevated expression level exclusively in the human lineage may indicate directional selection on gene expression in humans, potentially resulting in positive selection for particular regulatory variants (*Blekhman et al., 2008*; *Gilad et al., 2006*; *Perry et al., 2012*). However, it is often difficult to determine whether lineage-specific expression changes are due to inter-species environmental differences or to natural selection on certain regulatory variants.

A complementary approach to understanding gene regulation and associated selection pressures utilizes within-species variation to identify genetic variants that affect gene expression levels. Such variants are referred to as expression quantitative trait loci, or eQTLs. Overall, there is some evidence that eQTLs are evolving under weak negative selection. For example, the magnitude of eQTL effect size on expression levels is weakly anti-correlated with eQTL allele frequency (*Battle et al., 2014*; *Glassberg et al., 2019*). Additionally, the set of human genes associated with an eQTL (eGenes) tend to be slightly depleted for genes relevant to universally conserved and essential cellular processes (*Popadin et al., 2014*; *Tung et al., 2015*; *Ward and Gilad, 2019*), and for genes at the center of protein interaction networks (*Battle et al., 2014*; *Mähler et al., 2017*). These observations are consistent with negative selection purging strong regulatory variants within species, particularly variants that regulate the expression of genes whose precise regulation is essential.

The eQTL mapping approach allows us to connect genetic variants to the genes they regulate, and provides a mechanistic explanation for a portion of the heritability of gene expression phenotypes (*Price et al., 2011*; *Wright et al., 2014*). However, not all gene expression variation can be explained by genetic variation, and of the genetic contribution, only about 20% of heritability can be explained by locally acting variants (referred to as *cis* eQTLs; *Albert et al., 2018*; *Price et al., 2011*). We assume that the remaining 80% of heritable expression variation is determined by distally acting regulatory variants (*trans* eQTLs). However, because *trans* eQTLs can be located anywhere in the genome, they are difficult to pinpoint, even in studies with large sample sizes (*Battle et al., 2014*; *Westra et al., 2013*). Therefore, regarding the identification of genes undergoing stabilizing selection within or between closely related species, the insights gained by *cis* eQTL approaches may be limited.

A third approach to understanding the evolutionary forces on gene expression is to directly quantify the degree of total gene expression variation (i.e. the variability) within or between populations or species. The variability of gene expression in a population is the sum of variability introduced by local and distal genetic regulation, as well as other sources, including epigenetic (*Bashkeel et al., 2019*) and environmental effects. The variability in gene expression observed in a given population is the result of all of these effects as well as any technical variability that was introduced during experimental data collection. While variability measurements alone cannot help disentangle the genetic component of variability from the non-genetic component, we do expect that natural selection will minimize the regulatory variation of genes with dosage-sensitive functions. Fortunately, it is possible to obtain relatively stable measurements of population variability for every expressed gene using a moderate sample size of just tens of individuals (*de Jong et al., 2019*).

The quantification of regulatory variability may provide a window into distinct biological phenomena that are difficult to ascertain by studying inter-species differences in mean expression levels. For example, the adaptability of a population in response to new stresses may be a general function of gene expression variability, as has been demonstrated in yeast populations (*Bódi et al., 2017*; *Wang and Zhang, 2011*; *Zhang et al., 2009*). Furthermore, identification of hypervariable genes, regardless of the genetic or non-genetic source of variability, may help identify genes and pathways which confer differences in disease susceptibilities and treatment responses (*Ho et al., 2008*; *Knowles et al., 2018*; *Simonovsky et al., 2019*). Direct measurements of variability may therefore be a useful complement to the analysis of differences in mean gene expression levels or to eQTL mapping, and could contribute to a better understanding of gene expression evolution and complex traits.

A previous analysis of gene expression variability within and between human populations found higher regulatory variation in genes associated with disease susceptibility (*Li et al., 2010*). This study did not otherwise identify any particular functional classes or features of genes that showed significant inter-population differences in expression variability – although given the relative genetic similarity, short evolutionary timescale, and migrations between human populations, we might not expect different signatures of selection to be apparent. Nonetheless, this study found that regulatory variability across genes correlates with the levels of genetic variability at nearby loci (*Li et al., 2010*), thereby providing some measure of support to the notion that differences in regulatory variability between genes is at least partly genetically encoded.

We sought to better understand selection pressures on gene regulation variability by collecting gene expression data from humans and chimpanzees (*Pan troglodytes*). The availability of suitable samples from chimpanzees has notoriously been a limitation for comparative functional genomic

studies. Here, we were able to collect the largest population sample of chimpanzee primary tissues to date, allowing not only for sensitive assessment of differences in mean expression levels between species, but also differences in variability within species. To better isolate the genetic component of variability, we complemented the analysis of gene expression levels with a comparative eQTL mapping approach.

## Results

We analyzed RNA-seq data from postmortem primary heart tissue samples of 39 human and 39 chimpanzee individuals (*Figure 1—source data 1*). Data from 11 of these humans and 18 of the chimpanzees were previously collected in our lab and published (*Pavlovic et al., 2018*). We obtained additional human data from the GTEx consortium (post-mortem heart, left ventricle), filtering for high quality samples with sufficient read depth (*Figure 1A–B*, see Materials and methods for filtering criteria). To complement the published data from chimpanzees, we generated new RNA-seq data from primary heart samples of 21 additional chimpanzee individuals (see Materials and methods). Because the data were collected at different times, and some of the data from humans were collected in different labs (the GTEx data), our overall study design introduces clear technical batch effects that are partly confounded with species (*Figure 1B*). Our focus, however, is on inter-species comparisons of gene expression variability and eQTLs, which are estimated and identified using the within-species data. The batch effects are therefore expected to have only a minimal impact on the reported results.

To analyze the RNA-seq data, we employed a uniform processing pipeline that only considers reads mapping to human-chimpanzee orthologous exonic regions (*Pavlovic et al., 2018*). Using

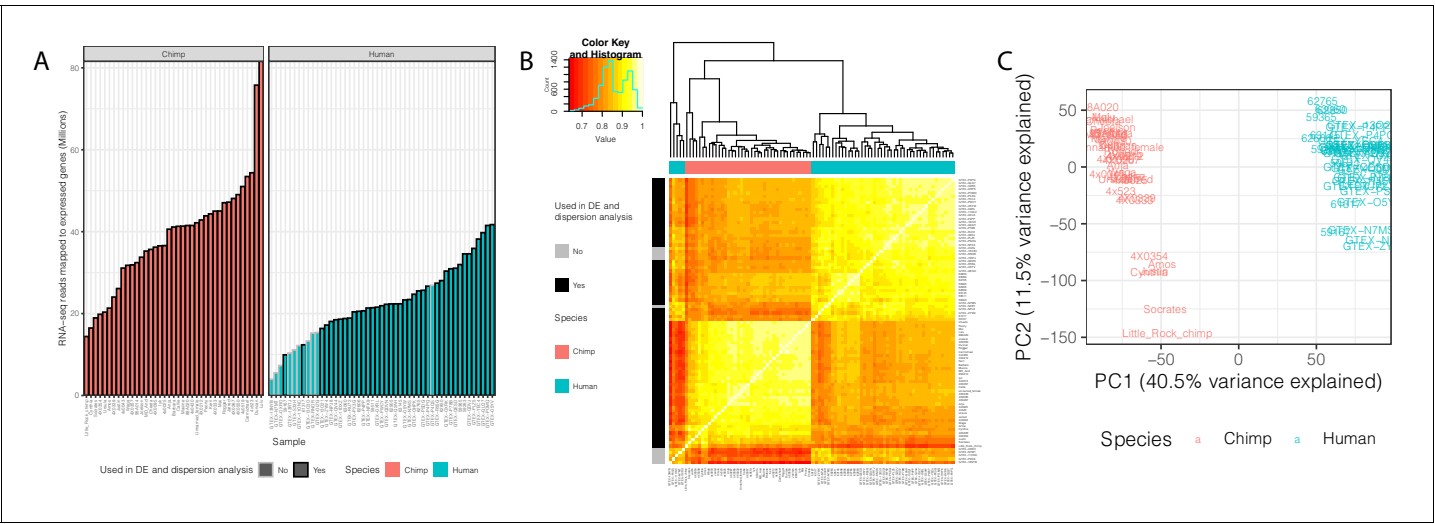

**Figure 1.** Summary of RNA-seq datasets. (**A**) Number of reads mapped to expressed genes for each sample. To obtain a balanced set of 39 samples per species, the 10 samples outlined in gray were excluded from differential expression and variability analysis. Nine of these samples are among the lowest read count samples sourced from GTEx. The remaining sample was excluded on the basis of inspection of (**B**) unsupervised hierarchal clustering of RNA-seq samples by Pearson correlation matrix of expression log(CPM). (**C**) Principal component analysis shows samples separating by species along the first principal component. Only samples used in DE and variability analyses are shown in (**C**). Source data for (**A**) and other metadata for RNA-seq datasets used in this study are in *Figure 1—source data 1*.

The online version of this article includes the following source data and figure supplement(s) for figure 1:

**Source data 1.** RNA-seq datasets used in this study.
**Figure supplement 1.** Effect size and significance of differential expression (DE) analysis.
**Figure supplement 1—source data 1.** Full DE results.
**Figure supplement 2.** Evaluation of the contribution of sample size and read depth to differential expression analysis between chimpanzee and human.
**Figure supplement 3.** Contribution of genetic relatedness to differential expression analysis between chimpanzee and human.
**Figure supplement 3—source data 1.** Kinship matrix of chimpanzees in this study.

these mapped reads, we estimated gene expression levels for each individual in each species. We excluded lowly expressed genes (see Materials and methods), and retained data from 13,432 expressed genes for further analysis. Principle component analysis (*Figure 1C*) and unsupervised clustering (*Figure 1B*) of the data show that, as expected, samples primarily separate by species, and not by source or batch, although as we pointed out, some technical batches are partly confounded with species in this study.

Using a linear model framework (see Materials and methods), we identified 8880 differentially expressed (DE) genes between species (FDR < 0.05), including 6,409 DE genes with effect sizes smaller than a twofold change (*Figure 1—figure supplement 1B*). As our comparative sample size is unusually large, we have an opportunity to comment on the robustness of observations that are made using study designs with smaller sample sizes, which are more typical for comparative studies in primates (*Khan et al., 2013*; *Pavlovic et al., 2018*; *Perry et al., 2012*). To do so, we subsampled our data and repeated the DE analysis for various sample sizes and read depths. To benchmark the results, we used the effect size estimates and DE gene classifications of the full dataset as an ad hoc gold standard reference.

As expected, we found that the number of individuals, not sequencing depth (within the range of 10M to >25M reads per sample), is the primary driver of power to detect inter-species differences in gene expression (*Figure 1—figure supplement 2F–O*). Our results indicate that while comparative studies with small sample sizes are underpowered, their reports of differentially expressed genes are generally robust. For example, when we used a sample size of only four chimpanzee individuals and four human individuals, we identified a median of 1,373 (869–2131 interquartile range across 100 resamples) DE genes (FDR < 0.05), or just 15% (10–24% IQR) of the genes identified as DE in the analysis of the full set of samples (*Figure 1—figure supplement 3A*). Furthermore, a study with only four individuals from each species is particularly underpowered to identify subtle inter-species differences (*Figure 1—figure supplement 2D*), only capturing a median of 5% (2–12% IQR) of the DE genes with effect sizes smaller than a twofold change. In addition, of the DE genes identified at this sample size, the estimated magnitude effect sizes are typically upwardly biased (*Figure 1—figure supplement 2E*) due to the winner's curse effects (*Göring et al., 2001*; *Ioannidis, 2008*), which particularly affect under-powered study designs (*Figure 1—figure supplement 2E*). However, even at this small sample size, the false positive rate associated with the classification of DE genes is well calibrated; when we used an FDR of 5% to classify genes as DE between the species, we empirically estimated a median of 2% (1–5% IQR) false discoveries based on the gold standard reference. Similar analyses for study designs with different sample sizes are available in *Figure 1—figure supplement 2*.

Considering that some of the chimpanzees we collected data from are first-degree relatives, we wondered if the presence of highly related chimpanzee samples plays a meaningful role in the inter-species DE comparisons. To examine that, we identified subsample sets of chimpanzees with varying degrees of inter-relatedness (*Figure 1—figure supplement 3A–B*). We then estimated the proportion of variance in gene expression that may be explained by relatedness. We found that technical factors, such as RNA extraction batch, play a larger role in explaining gene expression variance (*Figure 1—figure supplement 3C–D*) than the inter-relatedness. Furthermore, we repeated the DE analysis procedures with subsamples of only the inter-related chimpanzee individuals, and did not find any meaningful difference in number of DE genes or estimated false discovery rate compared to subsamples of less related individuals (*Figure 1—figure supplement 3E–F*). We conclude that the presence of inter-related samples plays a minimal role in altering interspecies DE analyses. The extrapolation of these analyses to guide interspecies study designs in relation to other factors such as tissue type and species, is not clear. Yet, we reason that tissue types that are more homogenous with less variability within species will naturally require fewer samples to detect similar effect sizes. Conversely, species comparisons that are more highly diverged, will likely have larger true effect size differences, and may require fewer samples to acquire meaningful results.

## Characterizing variability in gene expression

The main focus of our study was to assess and compare the inter-individual variability in gene expression between the two species. To do so, we estimated overdispersion for each gene in chimpanzees and humans separately (see Materials and methods). Briefly, we assumed that the measurement process is captured by Poisson sampling, and that true gene expression follows a Gamma

distribution. These assumptions are supported by theory (*Pachter, 2011*) and empirical data (*Marioni et al., 2008*). Accordingly, we used a negative binomial regression model to fit the RNA-seq data (22,23). Fitting the model to the data from each gene yields estimates of the mean and biological variance of gene expression, which result in the overdispersion of the observed read counts relative to a Poisson distribution.

Consistent with previous findings (*Ecker et al., 2017*; *Eling et al., 2018*; *Robinson et al., 2010*), we observed that overdispersion is strongly correlated with mean expression, due to reasons which may be technical or biological in nature. To understand the properties of gene expression variability that are independent of mean gene expression level, we regressed out the mean from the overdispersion estimates, to obtain a gene-wise mean-corrected summary of variability, hereafter referred to as 'dispersion' (*Figure 2A–B*). A dispersion greater than 0 indicates more variability than expected given the gene's expression level. Conceptually, our approach is similar to a method recently devised to identify differentially variable genes using single cell gene expression data (*Eling et al., 2018*). Using this approach, one can identify genes whose inter-individual variance in expression is different, irrespective of their mean or median expression levels. For example, while the genes *TERF2* and *SNORD14E* have similar median expression levels, *TERF2* has a lower dispersion than *SNORD14E,* in both species (*Figure 2B*). We used a bootstrap test (Materials and methods) to assess the stability of dispersion estimates and identified genes whose estimated dispersion is

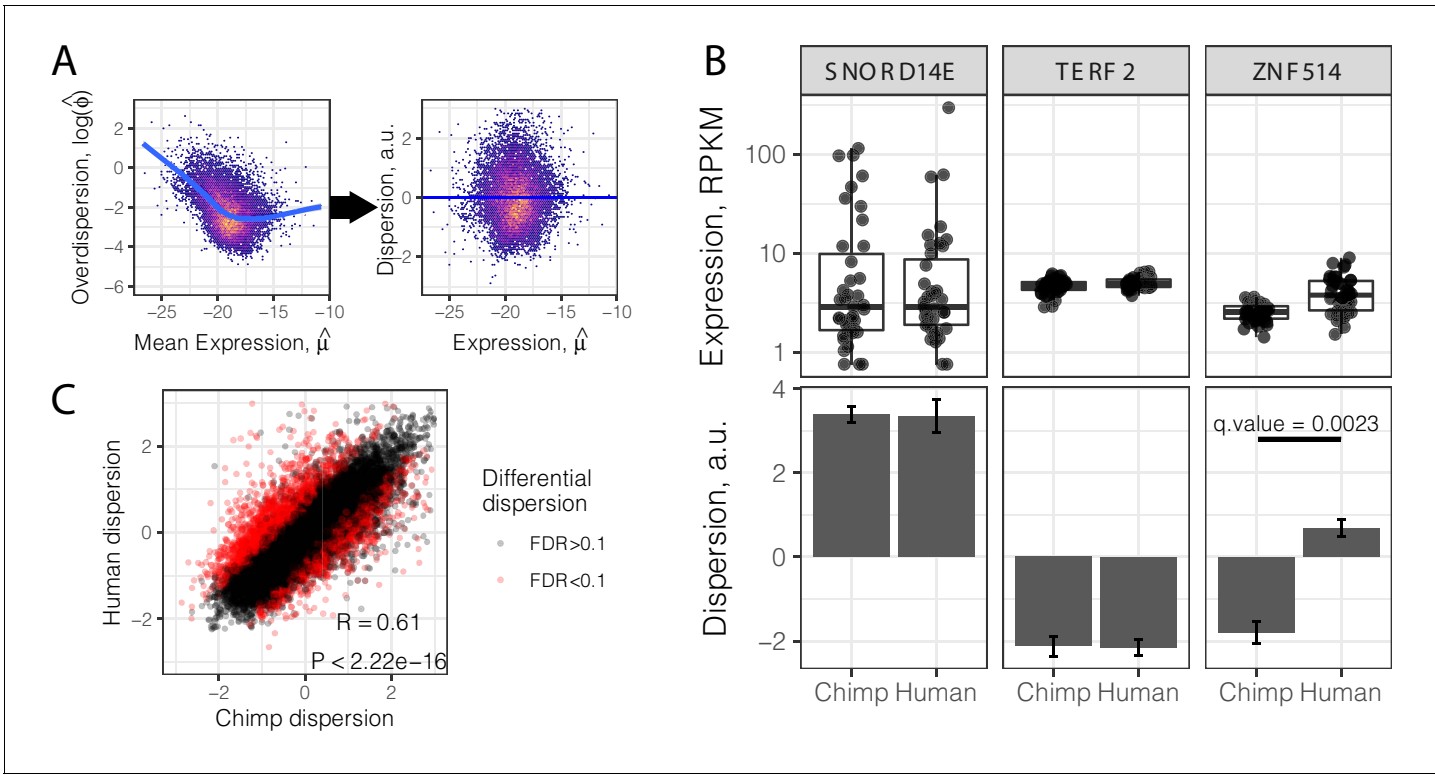

**Figure 2.** Gene variance independent of expression is correlated across species. (A) To estimate dispersion, RNA-seq counts for 39 human heart tissue samples were used to estimate gene-wise mean (μ) and overdispersion (φ) parameters. Across all genes, overdispersion is correlated with mean expression (left) in the hexbin scatter plot. We regressed out this correlation, using the residual of a LOESS fitted line (blue) as a metric (dispersion) of the variability of a gene's expression across a population relative to similarly expressed genes (a.u., arbitrary units). (B) Dispersion estimates and the underlying expression in each sample for three similarly expressed genes in human and chimpanzee. Error bars represent bootstrapped standard error. Q-value for *ZNF514* represents an estimate of FDR after genome-wide multiple hypothesis testing correction. (C) Dispersion estimates across all genes are correlated across human and chimpanzee, despite identification of thousands of differentially dispersed genes in red. R and p-value correspond to Pearson's correlation. Full dispersion estimates and differential dispersion results available as *Figure 2—source data 1*.

The online version of this article includes the following source data and figure supplement(s) for figure 2:

**Source data 1.** Gene-wise dispersion estimates and differential testing.
**Figure supplement 1.** Interspecies dispersion estimates are largely independent from interspecies mean expression changes.

significantly different between species. For example, *ZNF514* has higher dispersion in humans than chimpanzees, despite being similarly expressed in both species (*Figure 2B*). In total, we identified 2658 inter-species differentially dispersed genes (FDR < 0.1). Inter-species differences in mean expression levels cannot explain this finding, as differentially dispersed genes are generally not differentially expressed (*Figure 2—figure supplement 1*).

Despite the identification of thousands of differentially dispersed genes between the species, gene-wise dispersion estimates are well correlated in humans and chimpanzees overall (R = 0.60, *Figure 2C*), suggesting similar determinants of variability in the two species. We asked what other gene properties may be associated with the degree of gene expression variability. Specifically, we hypothesized that highly conserved essential genes, whose coding regions evolve under strong negative selection, would also be less variable in their expression across individuals. Indeed, when we ranked and grouped genes by degree of protein coding conservation (assessed by percent amino acid identity between human and chimpanzee; *Figure 3A*), or by the ratio of nonsynonymous to synonymous codon changes (dN/dS) across mammals (*Figure 3B*), we found that lower dispersion in expression levels is associated with higher protein coding conservation. This trend is most significant for genes in functional categories (Gene Ontology; GO categories) related to immune function

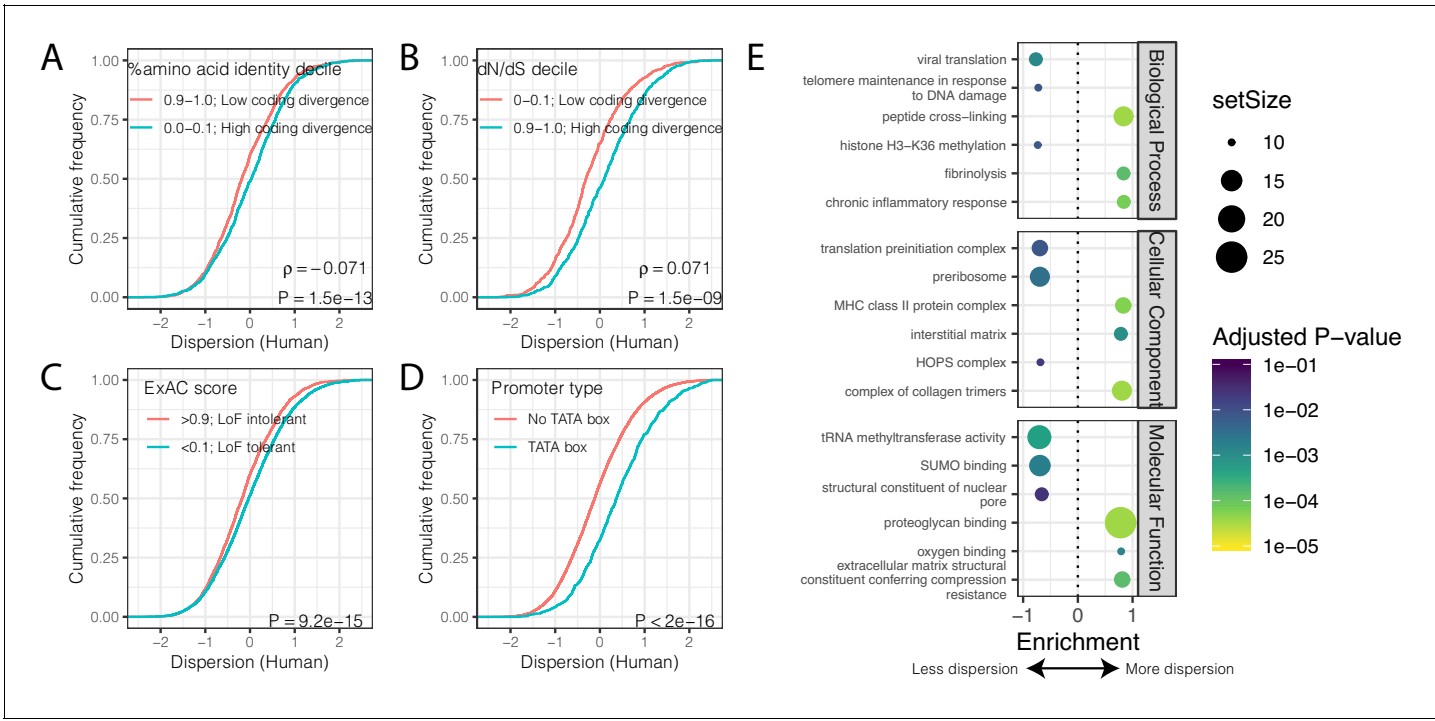

**Figure 3.** Gene features correlated with expression variability. (A) Protein coding genes with high coding divergence (defined by amino acid identity between chimpanzee and human) generally have higher variability than genes with low coding divergence. The distribution of dispersion estimates is plotted as the empirical cumulative distribution function (ECDF) for the top and bottom decile genes by percent identity. (B) Same as (A) but defining coding divergence based on ratio of non-synonymous to synonymous substitution rates (dN/dS) across mammals. (C) Loss-of-function tolerant (LoF tolerant) genes, defined by pLI score (*Lek et al., 2016*), generally have higher variability than loss-of-function intolerant (LoF intolerant) genes. (D) TATA box genes generally show higher variability. P-values and ρ correlation coefficient provided for (A) and (B) represent Spearman correlation across all quantiles, rather than just the upper and lower decile, which are plotted for similar visual interpretation as (C) and (D), where the P-values provided represent a two-sided Mann-Whitney U-test. (E) Gene set enrichment analysis (GSEA) of genes ordered by human dispersion estimates. Only the top and bottom three most enriched significant categories (Adjusted p-value<0.05) are shown for each ontology set for space. Full GSEA results available as *Figure 3—source data 1*.

The online version of this article includes the following source data and figure supplement(s) for figure 3:

**Source data 1.** Full GSEA results based on human dispersion levels.

**Figure supplement 1.** Correlation of coding conservation and dispersion across gene categories.

**Figure supplement 1—source data 1.** dN/dS correlation with dispersion by GO category.

**Figure supplement 2.** Gene features correlated with expression variability (chimpanzee).

(*Figure 3—figure supplement 1*), consistent with immune-related genes being targets of rapid evolution by diversifying selection pressures, which may increase dispersion. To further establish the relationship between evolutionary coding conservation and dispersion, we utilized the prevalence of genetic diversity data from humans to categorize genes as loss-of-function-tolerant (ExAC pLI score <0.1) or loss-of-function-intolerant (ExAC pLI score >0.9) (*Lek et al., 2016*). We found that the expression of loss-of-function-intolerant genes is associated with lower dispersion (*Figure 3C*). Unsurprisingly, these features of low dispersion genes similarly hold in chimpanzee (*Figure 3—figure supplement 2*) as in human (*Figure 3*).

Next, we asked what functional GO categories are enriched among the most or least dispersed genes. To do this, we ordered genes by their dispersion estimate in humans (*Figure 3E*) or chimpanzees (*Figure 3—figure supplement 1E*) and used gene set enrichment analysis (GSEA) (*Subramanian et al., 2005*) to identify GO categories enriched at the top or bottom of the list. We found that genes with housekeeping functions universal to all cell-types, such as genes related to transcription initiation and tRNA modification, are among the most enriched gene categories associated with low dispersion (*Figure 3E*, *Figure 3—figure supplement 1E*). Conversely, genes with high dispersion are enriched for categories like fibrinolysis (the breakdown of blood clots), oxygen sensing, and other cardiovascular related functions (*Figure 3E*, *Figure 3—figure supplement 1E*). Finally, consistent with previous reports (*de Jong et al., 2019*; *Hagai et al., 2018*; *Ravarani et al., 2016*), we found that genes with TATA boxes are associated with higher dispersion (*Figure 3D*), potentially because TATA boxes are associated with higher transcriptional noise at the molecular level (*Blake et al., 2006*; *Raser and O'Shea, 2004*; *Ravarani et al., 2016*). However, a possible technical explanation for this observation is that TATA box genes are enriched among cell-type-specific genes (*Schug et al., 2005*) and cell-type heterogeneity between individuals or samples could contribute to the observed dispersion.

To investigate the possibility that differences in cell composition between samples contribute to our observations of gene expression dispersion, we first asked whether dispersion is associated with the degree of cell-type-specificity of expression. We hypothesized that genes with high inter-individual dispersion are more likely to have cell-type-specific gene expression signatures in single-cell RNA-seq datasets of heart tissue. To examine this we turned to the *Tabula Muris* dataset, a comprehensive single-cell transcriptomics dataset which includes single-cell RNA-seq data from adult mouse heart tissue (*Tabula Muris Consortium et al., 2018*). Qualitatively, we observed that the most highly dispersed genes in our bulk chimpanzee and human samples have more cell-type-specific expression among the nine heart cell types identified in *Tabula Muris* (*Figure 4A*). Conversely, the lowest dispersed genes are more evenly expressed across cell types (*Figure 4B*). More generally, when we summarized the level of cell-type-specific expression for each gene as a single summary statistic, τ (*Kryuchkova-Mostacci and Robinson-Rechavi, 2016*; *Yanai et al., 2005*), we found that dispersion is strongly correlated with cell-type-specificity (R = 0.32, $p<2\times10^{-16}$; *Figure 4C*). Notably, the degree of cell-type-specificity is derived from data from mouse hearts, which may have diverged cell type expression profiles compared with primates. The correlation between cell-type-specificity and dispersion may therefore be downwardly biased as a result of error in estimating the true degree of cell-type-specificity of genes expressed primate hearts.

Given the correlation between dispersion and the extent of cell-type-specific regulation, we sought to estimate the proportions of different cell types amongst our bulk RNA-seq samples for both chimpanzee and human. We applied BayesPrism cell type deconvolution and expression estimation (*Chu and Danko, 2020*) to the bulk RNA-seq profiles using reference cell type profiles derived from *Tabula Muris* (Materials and methods). As expected, cell type proportions between chimpanzee and human hearts are qualitatively similar, although much inter-individual variation exists in both species for particular cell types, such as cardiac muscle cells and myofibroblasts (*Figure 4—figure supplement 1A*). Furthermore, using estimates of expression profiles within each cell type (*Figure 4—figure supplement 1C*), we calculated dispersion for human and chimpanzee within each cell type (*Figure 4—figure supplement 1D–E*). We found that the deconvoluted expression profiles for different cell types cluster tightly by cell type rather than species, as expected. Interestingly, the dispersion estimates cluster in a complex pattern that is more strongly influenced by species. This is consistent with the idea that genetic variation, which nearly completely segregates by species, plays a meaningful role is explaining dispersion when cell composition variation is corrected for. Put together, these analyses indicate that the levels of inter-individual dispersion we observed in

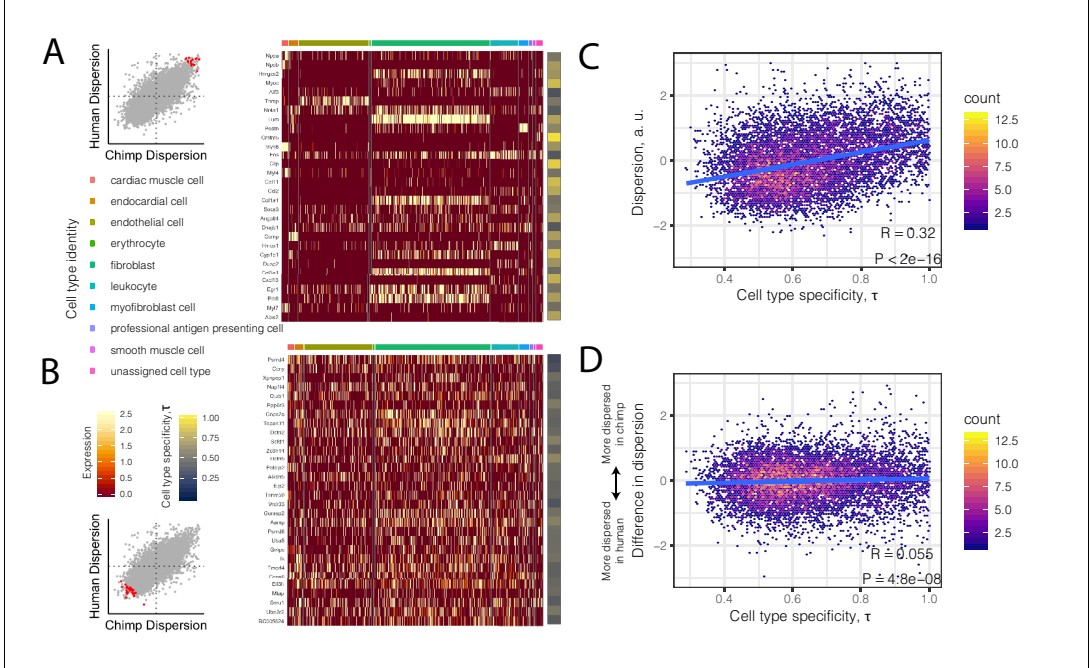

**Figure 4.** High dispersion genes are expressed in a cell-type-specific manner. (**A**) The top 30 most dispersed genes (the mean of the dispersion estimates between human and chimpanzee) are shown in a mouse heart single cell RNA-seq dataset (*Tabula Muris Consortium et al., 2018*). Scatterplot inset on the left shows the dispersion estimate of the 30 genes in chimpanzee and human (red points) compared to all other genes (gray). Each row in the heatmap is a gene. Each column is a single cell, grouped by cell type (colors at top of columns). Normalized expression is colored in the body of the heatmap. The τ statistic, colored on the right of the heatmap, summarizes for each gene how cell-type-specific the gene is, ranging from 0 (equally expressed among all cell types) to 1 (expressed exclusively in a single cell type). (**B**) The same as (**A**) but for the bottom 30 most dispersed genes. (**C**) Across all genes, a hexbin scatterplot shows the correlation between cell-type-specificity (τ) estimated from mouse single-cell RNA-seq data, and dispersion (mean of human and chimpanzee dispersion for each gene) estimated from the bulk RNA-seq data. (**D**) The difference in dispersion between chimpanzee and human is only weakly correlated with cell-type-specificity. R and p-value for (**C**) and (**D**) represent Pearson's correlation.

The online version of this article includes the following source data and figure supplement(s) for figure 4:

**Figure supplement 1.** Expression and dispersion estimates that correct for cell type composition.
**Figure supplement 1—source data 1.** Cell-type-specific expression and dispersion estimates.
**Figure supplement 2.** Cell type heterogeneity has a strong non-technical, individual-specific component.
**Figure supplement 3.** GSEA of genes with different levels of dispersion between species.
**Figure supplement 3—source data 1.** Full GSEA results based on interspecies dispersion differences.

humans and chimpanzees, and the high similarity in dispersion observed across species, are driven genetic differences as well as cell type heterogeneity between samples. The cell type heterogeneity across samples may be partly due to both technical differences between sample preparations, as well as biological differences between individuals.

To investigate the extent to which the cell composition differences that drive dispersion are biologically relevant, as opposed to technical differences in tissue dissection and sample preparation, we turned to cell deconvolution profiles of samples from anatomically different heart samples from matched individuals from GTEx (*Donovan et al., 2020*). We reasoned that if intentionally anatomically different heart sections (left ventricle, versus atrial appendage) from the same individual correlate better than matched tissue samples from different individuals, then the cell type composition differences across our chimpanzee samples are also likely driven by individual level differences, rather than technical differences in sample acquisition.

We found that the estimated fraction of cardiac muscle cells, the most common cell type in both tissue sections, is highly correlated between atrial appendage and left ventricle samples from matched individuals (*Figure 4—figure supplement 2A*). We used a linear mixed model (*Hoffman and Schadt, 2016*) to quantify the contribution of individual level versus tissue level factors to explain cell type composition estimates, and found that the individual level factor generally

explains more variance (*Figure 4—figure supplement 2B*). We interpret this as strong evidence that the sample-to-sample differences captured by our dispersion estimates are driven largely by true differences between individuals, rather than random technical differences in sample acquisition and dissection.

Having more confidence that the dispersion estimates reflect true biological variability, we sought to characterize the differences in dispersion between chimpanzee and human. Importantly, we find that the interspecies *difference* in dispersion is much less likely to be driven by cell type heterogeneity, although there is a relatively small but significant correlation with $\tau$ (R = 0.07, p=2.2×10$^{-12}$; *Figure 4D*). We next asked what gene categories are enriched among differentially dispersed genes.

We performed GSEA (*Subramanian et al., 2005*), ranking genes by the polarized significance level of the chimpanzee-human difference in dispersion estimates. We found that genes more variable in human are enriched for mitotic regulators (*Figure 4—figure supplement 3*). Given evidence that ischemia may induce mitosis in adult mammalian cardiac cells (*Kajstura et al., 1998*; *Kimura et al., 2017*; *Nakada et al., 2017*), the enrichment for mitotic regulators may reflect the highly variable life histories in GTEx samples, a large fraction of which are sourced from organ donors with ischemic cardiovascular disease. Conversely, we found that genes related to immune function are more variable in chimpanzee. We note that some of our chimpanzee individuals were sourced from laboratory settings and have been challenged with HBV or HCV viral infections. However, our GSEA results are robust to the exclusion of these samples (*Figure 4—figure supplement 3—source data 1*, *Supplementary file 1*).

## Within-species genetic variation contributes to inter-species differences in variability

Although potentially technical in nature, inter-individual differences in cellular composition provide a partial explanation for our observation of similar dispersion estimates across species. We looked for evidence that genetic diversity also drives dispersion. We asked whether inter-species differences in dispersion, which are much less likely to be explained by cellular composition, are associated with corresponding differences in selection pressures between the species. We reasoned that if inter-species differences in dispersion are partially driven by inter-species differences in selection pressures, we may see differences between the species in genetic signatures that are consistent with natural selection near the differentially dispersed genes. More specifically, we expect that genes with particularly low dispersion in human compared to chimpanzee will also display more constraint at the coding level in human than in chimpanzee. To assess this, we analyzed genotype data from our chimpanzee and human cohorts. We obtained human genotype data from the GTEx consortium, and chimpanzee data by performing high-coverage whole genome sequencing on the 39 chimpanzee samples used in this study (>30X genome coverage obtained in all samples, *Supplementary file 2*); these data represent roughly a 50% increase in the number of high-coverage *Pan troglodytes* genomes currently available (*de Manuel et al., 2016*). We used the sequencing data to identify nearly 2.9 million novel chimpanzee SNPs with a minor allele frequency (MAF) greater than 10%.

Using the genotype data, we asked if differences in expression dispersion are associated with differences in evolutionary constraint on protein coding regions in the human and chimpanzee lineages. For each gene, we calculated the ratio of non-synonymous polymorphisms ($P_n$) scaled to synonymous polymorphisms ($P_s$) within each species. This $P_n/P_s$ metric may be used to assess purifying and diversifying selection pressures acting on coding sequences *within* species (*Fuller et al., 2015*; *Huguet et al., 2014*; *Tanaka and Nei, 1989*), which thus may correspond to our measurements of within-species expression dispersion. In contrast, the more often-utilized dN/dS metric is based on fixed coding differences between species and therefore not well suited to identify the signatures of selection that uniquely confine variability *within* a species. As expected, we found that loss-of-function tolerant genes have higher $P_n/P_s$ than loss-of-function intolerant genes (*Figure 5A*). When we compared $P_n/P_s$ between humans and chimpanzees, we found that the inter-species ratio of $P_n/P_s$ is positively correlated with the difference in gene expression dispersion between species (*Figure 5B*). That is, on average, genes with higher dispersion in chimpanzees than in humans have a higher abundance of non-synonymous polymorphisms ($P_n/P_s$) in chimpanzees compared to the orthologous genes in humans. This suggests that inter-species differences in selection pressures at polymorphic loci play a role in the observed inter-species differences in expression dispersion.

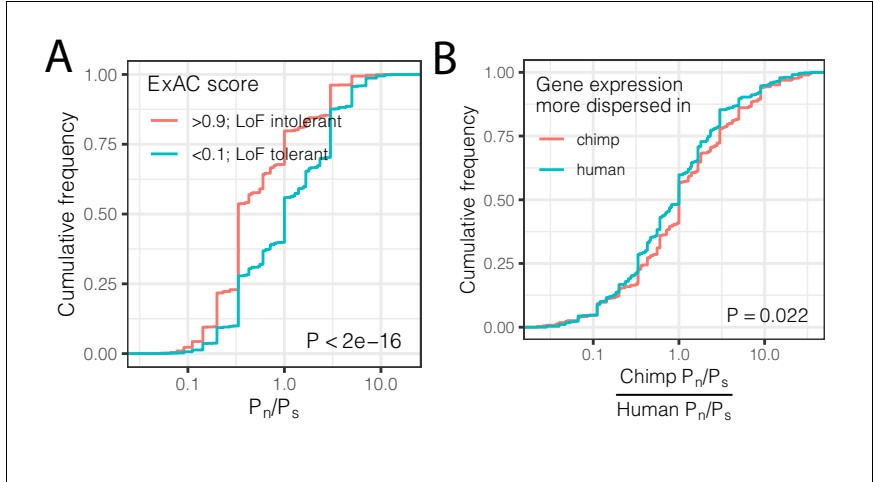

**Figure 5.** Interspecies differences in dispersion correlate to interspecies differences in coding constraint. (**A**) The number of nonsynonymous polymorphisms scaled to synonymous polymorphisms ($P_n/P_s$) for each gene was calculated in GTEx human population. Loss-of-function tolerant (LoF intolerant) genes, defined by pLI score (*Lek et al., 2016*), generally have higher $P_n/P_s$ than loss of function tolerant (LoF tolerant) genes, as shown in ECDF plot. (**B**) $P_n/P_s$ was calculated for both human and chimpanzee. The distribution of the chimpanzee $P_n/P_s$ to human $P_n/P_s$ ratio is plotted as an ECDF, grouped by whether the gene has higher dispersion estimate in chimpanzee than in human. p-Value indicates a Mann Whitney U-test. Gene-wise $P_n/P_s$ statistics available in *Figure 5—source data 1*.

The online version of this article includes the following source data for figure 5:

**Source data 1.** Gene-wise Pn/Ps statistics for chimpanzee and human.

## Genes with eQTLs are shared across species more often than expected by chance

The correlation between the degree of evolutionary constraint on coding sequences and dispersion of expression at the gene level suggests that differences in cellular composition are not the only explanation for differences in dispersion. Motivated by this notion, we searched for further evidence for genetic regulation of dispersion by identifying genes associated with eQTLs - eGenes - in both humans and chimpanzees. In humans, we obtained a list of 11,682 heart left ventricle eGenes identified by the GTEx consortium (with a sample size of 386 individuals). In chimpanzee, we used the genotype and expression data we collected to map *cis* eQTL SNPs within 250 kb of each of the 13,545 expressed genes. We included 10 principal components as covariates (*Figure 6—figure supplement 1*) and accounted for genetic relatedness between individuals and population structure, which we inferred from the genotype data (Materials and methods and *Figure 6—figure supplement 2*). Because gene expression principle components are correlated with cellular heterogeneity (*Figure 4—figure supplement 1B*), their inclusion as covariates in the eQTL linear modeling helps correct for this heterogeneity and additional unobserved technical sources of variation. Using this approach, we identified 310 eGenes in chimpanzee hearts (FDR < 0.1; *Figure 6—figure supplement 1A–B*, *Supplementary file 3*). Consistent with previous eQTL studies in primates (*Jasinska et al., 2017*; *Pickrell et al., 2010*; *Tung et al., 2015*; *Veyrieras et al., 2008*), we found that the chimpanzee eQTL SNPs are enriched near transcription start sites (*Figure 6—figure supplement 1C*).

We considered the overlap of eGenes in humans and chimpanzees. When considering only one-to-one orthologs tested for eGenes in both species, there is no significant overlap of eGenes in the two species (Odds Ratio = 1.03, p=0.46, hypergeometric test, *Figure 6—figure supplement 3A*). However, this comparison is affected by the substantial difference in power to detect eQTLs between the large human sample used in GTEx and the relatively small chimpanzee sample we collected. To address this, we iteratively subsampled the human cohort to sample sizes comparable to our chimpanzee cohort, and re-mapped human eQTLs. When we compared lists of eGenes identified in humans and chimpanzees using similar sample sizes, we found a greater overlap of eGenes in the two species than expected by chance (*Figure 6—figure supplement 3B*). This observation is

robust even if we use the eQTL results of the full GTEx dataset, as long as we perform the comparison by using the largest-effect eQTLs (those that can be identified as significant even in smaller sample sizes). Specifically, of the top 500 significant GTEx eGenes (*Figure 6A*, *Figure 6—figure supplement 3C*), 21 are also found to be eGenes in chimpanzee (FDR < 0.1), a significant enrichment (Odds Ratio = 2.05, p=0.003, hypergeometric test). Importantly, we found that species-specific eGenes have higher species-specific dispersion (*Figure 6B*). That is, eGenes identified in chimpanzees but not humans have higher dispersion in chimpanzees and vice versa (p=$3.7\times10^{-11}$; *Figure 6B*). Furthermore, within each species, eGenes tend to have higher dispersion than non-eGenes (*Figure 6—figure supplement 4*). These observations are consistent with a genetic contribution to inter-species differences in gene expression dispersion. Furthermore, the observation that more eGenes are shared among humans and chimpanzees than expected by chance suggests that the regulation of these genes evolves under less evolutionary constraint.

## Shared and species-specific eGenes may evolve under different selection pressures

To further examine whether shared and species-specific eGenes may evolve under different selection pressures than non-eGenes, we examined other indicators of selection for each of these eGene groups. We found that eGenes have higher inter-species differences in expression levels than non-eGenes (*Figure 7A*), and eGenes identified in both species have even larger differences in mean expression levels between species than eGenes identified only in chimpanzees or humans. These

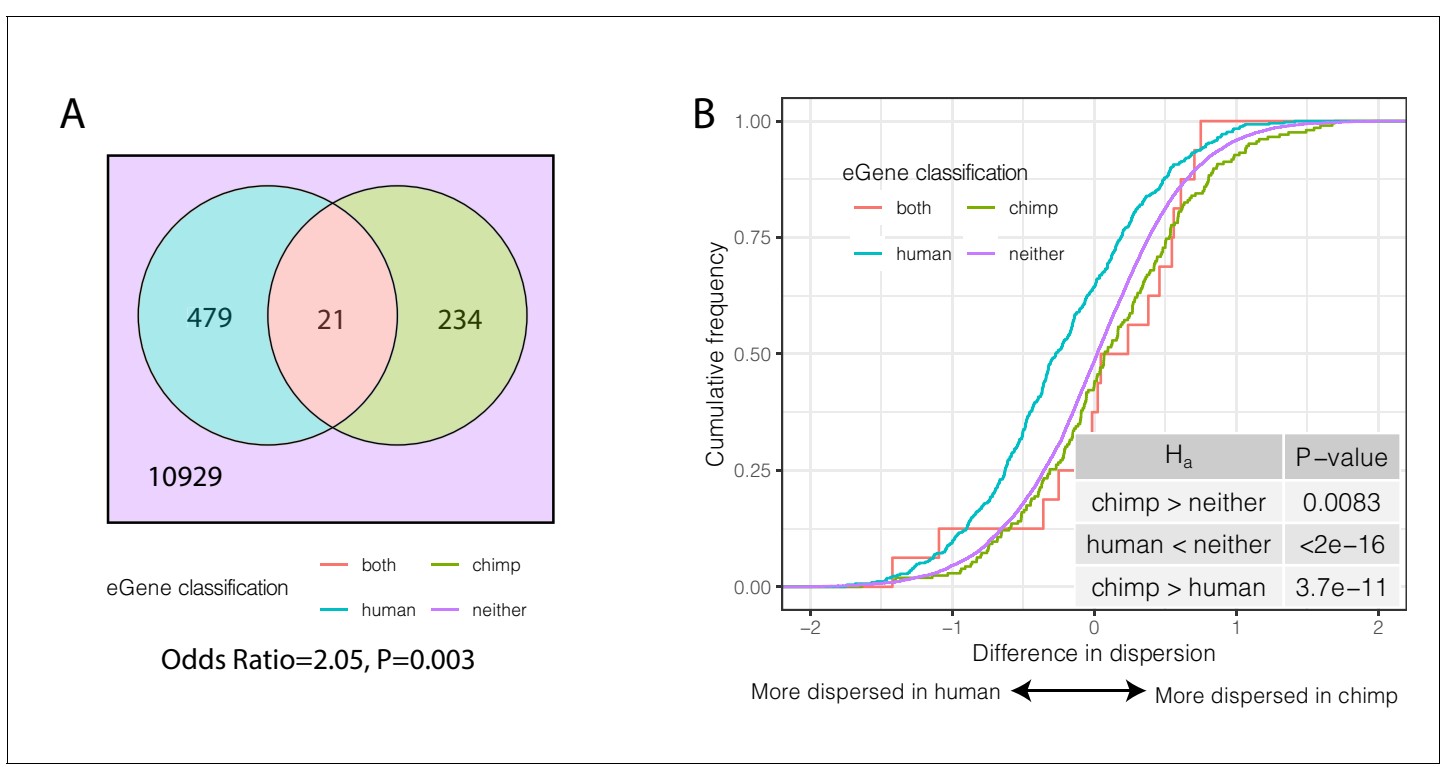

**Figure 6.** Species-sharing and dispersion of eGenes. (**A**) eGenes were classified by a 10% FDR threshold in chimpanzee and considering only the top 500 eGenes by FDR in human (GTEx). (**B**) ECDF of the difference in dispersion of genes between chimpanzee and human. Chimpanzee-specific eGenes are more dispersed in chimpanzee; human-specific eGenes are more dispersed in human. p-Values provided for one-sided Mann-Whitney U-test with the noted alternative hypothesis.

The online version of this article includes the following source data and figure supplement(s) for figure 6:

**Figure supplement 1.** eQTL mapping in chimpanzee samples.

**Figure supplement 2.** Population structure and relatedness of chimpanzee cohort based on whole genome sequencing genotyping.

**Figure supplement 2—source data 1.** Admixture group membership of chimpanzees in this study.

**Figure supplement 3.** Significant overlap of eGenes between chimpanzee and human is observed when studies are similarly powered.

**Figure supplement 4.** eGenes have higher dispersion than non-eGenes.

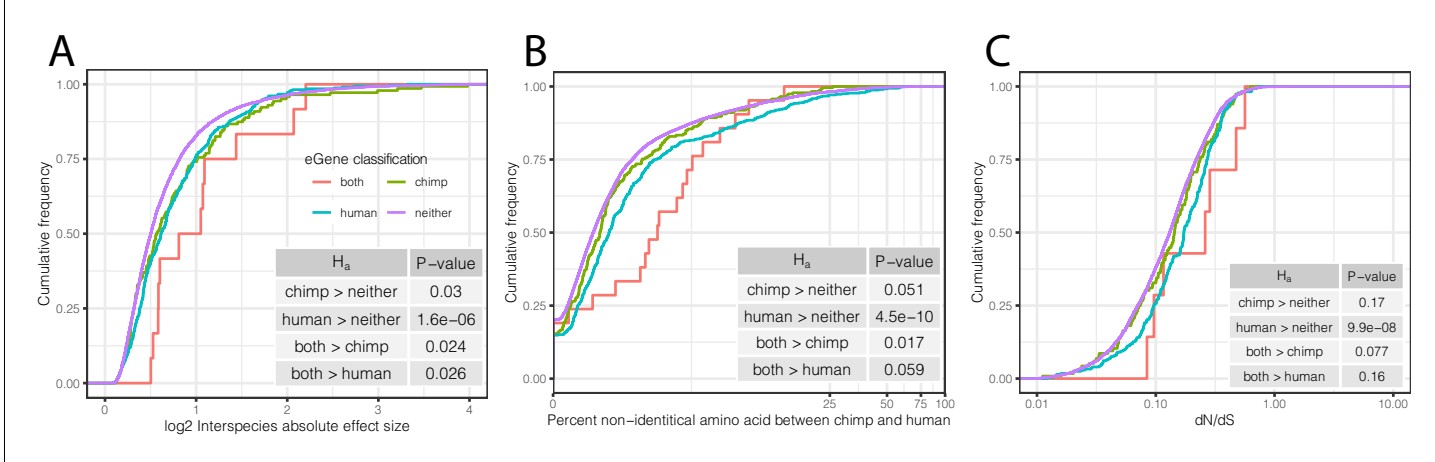

**Figure 7.** Characteristics of eGenes are consistent with less constraint on eGene expression. (**A**) eGenes are more differentially expressed between species than non-eGenes, with eGenes detected in both species being even more differentially expressed. The distribution of the inter-species differential expression effect size is plotted for each eGene group as an ECDF. (**B**) eGenes are more diverged at amino acid level than non-eGenes. (**C**) Human-specific eGenes and shared eGenes are more divergent than expected under neutrality. The analogous test for chimpanzee-specific eGenes displayed a shift that was not statistically significant, although our classification of eGenes in chimpanzee may be underpowered. p-Values provided for one-sided Mann-Whitney U-tests with the noted alternative hypothesis.

The online version of this article includes the following source data and figure supplement(s) for figure 7:

**Figure supplement 1.** Gene categories of human-chimpanzee shared and chimpanzee-specific eGenes.
**Figure supplement 1—source data 1.** Full GO enrichment results of species-shared eGenes.
**Figure supplement 1—source data 2.** Full GO enrichment results of chimpanzee-specific eGenes.

observations are consistent with the notion that the regulation of genes associated with eQTLs tend to evolve under less evolutionary constraint. Furthermore, eGenes tend to have lower levels of coding conservation in both species, as measured by amino acid identity between human and chimpanzee (*Figure 7B*) or dN/dS across mammals (*Figure 7C*).

We next performed GO enrichment analysis (hypergeometric test) to ask which functional classes are identified as eGenes in both human and chimpanzee. We reasoned that the 21 genes identified as shared eGenes may have high levels of genetically regulated variability, which may indicate expression evolution at a neutral rate or faster. We found that these genes are strongly enriched for immune response genes, including major histocompatibility complex (MHC) genes (*Figure 7—figure supplement 1A*). This observation is consistent with previous reports that immune genes evolve under strong directional and balancing selection pressures across vertebrates, in part to respond to ever-evolving pathogen challenges (*Ejsmond and Radwan, 2015*; *Hagai et al., 2018*; *Lam et al., 2017*; *Shultz and Sackton, 2019*).

Given the evidence that highly dispersed genes and eGenes are associated with relaxed evolutionary constraint, we next asked which gene classes are enriched among chimpanzee-specific eGenes. This set of genes may be subject to stronger stabilizing selection in the human lineage. We approached this question by considering all eGenes discovered in the full GTEx dataset (FDR < 0.1) as human eGenes, as this is the most stringent way to classify eGenes as chimpanzee-specific. We identified 148 chimpanzee-specific eGenes, which we found to be significantly enriched for transcriptional regulation terms (*Figure 7—figure supplement 1B*).

## Effects of trans-species polymorphisms on gene expression

Genetically driven variability in gene expression may also arise due to overdominant or frequency-dependent selection on gene regulation, which maintains polymorphisms over evolutionary time through balancing selection (*Croze et al., 2016*; *Těšický and Vinkler, 2015*). MHC and other immune genes are well known targets of these modes of selection, as host immune systems are under constant evolutionary pressure to diversify in response to quickly evolving pathogens. As

such, these genes sometimes contain trans-species polymorphisms maintained through evolutionary time by balancing selection (*Croze et al., 2016*; *Těšický and Vinkler, 2015*).

A previous study identified 125 trans-species polymorphic haplotypes outside of the MHC region that are shared between chimpanzee and human, all but two of which are in noncoding regions (*Leffler et al., 2013*). Whether these polymorphisms are maintained by balancing selection because of their potential for regulatory effects on gene expression is not clear. We found that the set of genes nearest to these trans-species SNPs have higher median levels of dispersion than distal genes, though the effect is small and may be due to chance (p=0.053, *Figure 8—figure supplement 1*). If these trans-species SNPs have conserved regulatory activity, which diversifies the expression levels of nearby genes, we would additionally expect to see similar eQTL effects in both human and chimpanzee. To test this, we remapped eQTLs for these SNPs in both species with a uniform pipeline (see Methods). We detected 37/192 trans-species SNPs with a clear eQTL signal (FDR < 0.1) in the well-powered human dataset, such as rs257899, which associates with *SLC27A6* expression. However, we did not identify any significant *cis* eQTL activity for this SNP in chimpanzee (*Figure 8A*). More generally, we did not find any inter-species correlation of regulatory effect size estimates among the 12 trans-species haplotypes with an eQTL in human (FDR < 0.1; *Figure 8B*) that were also tested in chimpanzee (Methods). Considering all trans-species SNPs, we did not find any evidence for their regulatory effects in chimpanzee, compared to a set of control SNPs (*Figure 8C*). While we found these SNPs to have measurable regulatory activity in the human dataset, it was not significantly different than that of control SNPs (*Figure 8D*).

Finally, we asked whether these SNPs may be under selection due to eQTL effects in tissues other than heart. To this end, we identified the most significant eQTL P-value for each of these SNPs across all GTEx tissues and found that these eQTL effects are not statistically different from that of control SNPs (*Figure 8—figure supplement 2A*). Furthermore, the tissues where the eQTL minimum p-value was identified are similar to those of control SNPs (*Figure 8—figure supplement 2B*), suggesting these trans-species SNPs in general do not have specific regulatory activity in any particular tissue. In summary, we found no compelling evidence that these trans-species polymorphisms have strong regulatory activity in either species.

## Discussion

We set out to understand the properties that are associated with different levels of gene expression variability in human and chimpanzee populations. Because we know that gene expression variation in humans is often associated with genetic variation (in the form of eQTLs), we hypothesized that the degree of population variability in gene expression levels may reflect the evolutionary constraint on gene regulation. We reasoned that a comparison of regulatory variation in humans and chimpanzees, and a comparative eQTL study, may provide evidence to support said hypothesis and further identify inter-species similarities and differences in the selective pressures on gene expression.

We found that inter-individual expression variability is highly correlated in humans and chimpanzees. At first glance, this seems to support the notion that regulatory variation evolves under similar selective pressures in both species. However, we were unable to exclude a technical explanation for this observation. It was difficult to disentangle the genetic and non-genetic contributions of this variability because we used primary tissue samples that include multiple cell types. We found that, across genes, cell type heterogeneity is a major driver of the degree to which gene expression varies in the population. Because orthologous genes in human and chimpanzee are expected to have similar expression patterns, this finding can potentially explain the observation of high correlation in expression dispersion in the two species. Though this technical explanation may be intuitive, the degree of the association between population variability and the cellular specificity of gene expression may have been overlooked without the use of single-cell data.

Cell type heterogeneity has likely affected previous comparative studies of gene regulation that used primary tissue samples, including studies from our own lab. We and others have commented on this property of primary tissue comparisons in the past (*Avila Cobos et al., 2018*; *Blekhman et al., 2008*; *Newman et al., 2015*; *Selewa et al., 2020*), but without single cell data it was impossible to effectively assess the magnitude of this effect. Our findings further underscore the need for single cell measurements to disentangle sources of variation in bulk RNA-seq data, especially from primary tissues. Future work examining population variability should take this into

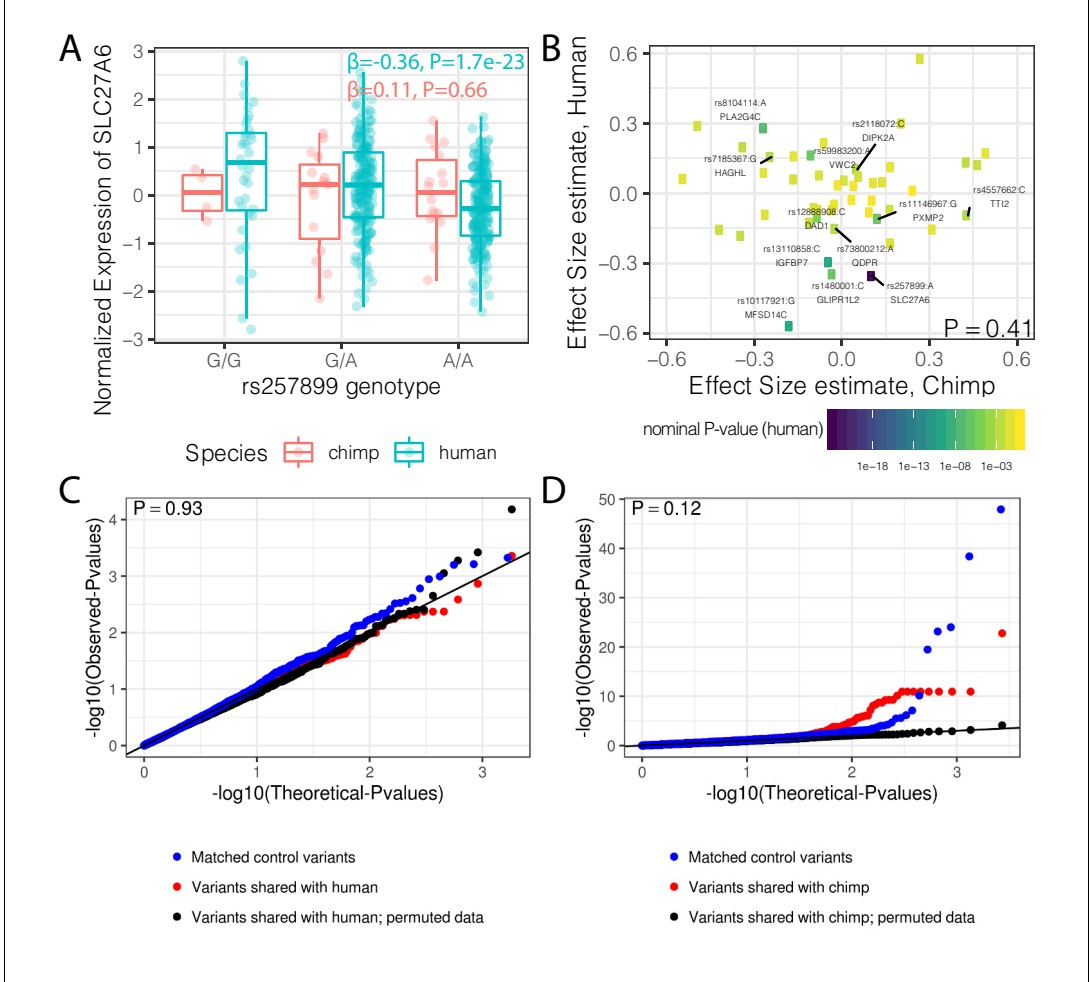

**Figure 8.** Trans-species polymorphisms do not detectably regulate gene expression. (**A**) Boxplot of *SLC27A6* expression stratified by species and genotype of the trans-species SNP rs257899, the most significant eQTL of the trans-species polymorphisms tested in human heart (left ventricle) in GTEx. eQTL effect size estimates (β) and nominal p-values are provided. (**B**) There is not a general correlation of gene regulation effects of trans-species SNPs between chimpanzee and human. For each trans-species polymorphic region previously identified (*Leffler et al., 2013*), the most significant SNP:gene pair in human is shown with the effect size estimate in both human and chimpanzee. Only the 48 regions where the strongest human SNP:gene association was also a one-to-one ortholog and the same SNP:gene pair was also tested in chimpanzee were plotted. Labeled SNP: gene pairs indicate FDR < 0.1 in human. One sided P-value provided for Pearson correlation, under the alternative hypothesis that effect sizes should be positively correlated between species. Only SNP:gene pairs FDR < 0.1 in human were considered for this test. (**C**) The trans-species polymorphisms do not have detectable *cis* eQTL activity in chimpanzee. QQ-plot of p-values of *cis* eQTL activity of the trans-species polymorphisms, compared to a sample permutation control, and to a control set of SNPs. (**D**) Same as (**C**) but testing *cis* eQTL activity in human. p-Values provided for (**C**) and (**D**) represent one-sided Mann-Whitney U-test with the alternative hypothesis that trans-species polymorphisms have smaller *cis* eQTL p-values than the control SNPs.

The online version of this article includes the following figure supplement(s) for figure 8:

**Figure supplement 1.** Genes closest to trans-species polymorphisms exhibit similar dispersion.

**Figure supplement 2.** Trans-species polymorphisms do not regulate gene expression differently than control SNPs across all GTEx tissues.

account, possibly by collecting single cell data, to separate cell-type heterogeneity from heterogeneity within a particular cell type. It is important to account for cellular composition not only in comparative studies of variation in gene expression, but also in studies that focus on a single species. Indeed, cellular heterogeneity may itself have a genetic component, which will be confounded with regulatory differences within a cell type (*Donovan et al., 2020*; *Marderstein et al., 2020*).

Notwithstanding these complications, our observations do indicate that natural selection has played a role in shaping inter-species similarities and differences in gene expression variability.

Differences in cellular composition cannot explain the observed correlation between dispersion and measures of nucleotide divergence and diversity. This was the first observation in our study that provided some measure of support for the hypothesis that regulatory variation may be a genetic trait. Although the inference of selection at the genomic level indicates selection on coding regions (not gene regulation), the correlation with the degree of variation in expression suggests that the regulation of functionally important genes is also a selected trait.

Encouraged by this finding, we were able to find more evidence to support our hypothesis by carrying out a comparative eQTL analysis. We identified eQTLs in humans by using the GTEx data, which sampled hundreds of individuals. In chimpanzee, we identified eQTLs by using our 39 samples. It is quite difficult to obtain chimpanzee primary tissue samples and although this sample size is modest, it is probably the largest population of chimpanzee primary tissue samples ever reported. The difference in sample size between the human and chimpanzee eQTL discovery panels means that observations of human-specific eGenes are quite expected and can often be explained by the fact that we have more power to detect eQTLs in humans. In contrast, the observation of chimpanzee-specific eGenes is quite meaningful because it is much less likely that the human sample was underpowered to detect eQTLs for said genes if they existed.

With that in mind, our observation that species-specific eGenes have higher variability in the species in which the eQTL was detected is significant, because it directly points to a genetic basis for differences in expression dispersion between humans and chimpanzees. This observation is also consistent with the notion that genetic variation within species contributes to the overall inter-species divergence in gene regulation. This is a critical piece of evidence supporting our hypothesis, although we acknowledge that under some complicated scenarios, one could evoke cellular composition as a potential explanation for this observation as well. We argue, however, that this would require inter-species differences in cellular composition to segregate with dozens of genotypes in just one species, and for these genotypes to appear as *cis* eQTLs for genes that have higher divergence due to cellular composition. The requirement for the genotypes to correlate with the difference in cellular composition while also being in proximity to the specific set of eGenes that would satisfy the divergence requirement is extremely unlikely, although we cannot offer empirical data to entirely exclude this possibility. That said, we also observed stronger signals of coding selection for non-eGenes than eGenes in both species, further suggesting selection on the eGenes themselves (this is not expected if our observations are to be explained by differences in cellular composition).

Collectively, our observations suggest that, across species, eQTLs may be a subtle indicator of dosage insensitivity for relatively neutrally evolving genes. Thus, we believe that an inter-species analysis of population variability in gene expression may be a relatively simple way to complement existing methods to assess differences in selection pressures between lineages. Interestingly, the eGenes we identified only in chimpanzee, but not in human, are slightly enriched for transcription regulatory processes, reminiscent of a previous observation that transcription factors seem to be enriched among the genes positively selected for in the human lineage (*Blekhman et al., 2008*; *Gilad et al., 2006*).

In both human and chimpanzee, we found that genes involved in immune response are among the most variably expressed and strongly enriched among species-shared eGenes. This is consistent with a body of literature (*Croze et al., 2016*; *Ejsmond and Radwan, 2015*; *Hagai et al., 2018*; *Shultz and Sackton, 2019*) that points to diversifying selective pressure on immune related cell-surface receptor genes to identify and combat diverse and ever-evolving pathogens. A prime example of this is the abundance of MHC complex genes among the most variable genes in both species, and strongly enriched in the subset of shared eGenes. However, the extent to which quantitative regulation of gene expression is functionally important to pathogen defense and thus the target of selection is unclear. Alternatively, these regulatory variants may be hitchhiking with functionally important and tightly linked coding variants under strong positive or balancing selection (*D'Antonio et al., 2019*; *Meyer et al., 2018*; *Shiina et al., 2009*). If non-coding trans-species polymorphisms are targets of long lived balancing selection on gene regulation (*Johnsen et al., 2009*; *Leffler et al., 2013*), we expect these polymorphisms to display similar eQTL effects in both species. However, we failed to identify generally conserved regulatory effects of trans-species polymorphisms. Although we found clear instances of regulatory effects from some of these trans-species SNPs in humans, we note that this human dataset is well powered enough to detect similar regulatory effects even from random control SNPs. The cumulative regulatory effects of trans-species

polymorphisms are not significantly different than the control SNPs. We acknowledge that we may be underpowered to detect subtle conserved regulatory effects in chimpanzee. Moreover, some of these trans-species polymorphisms may have important regulatory functions; albeit this function may also be tissue-, cell-type-, or context-dependent and not present in our assessment of heart and other GTEx bulk tissues.

In summary, we performed a comparative assessment of expression variability and eQTL mapping and found signatures of stabilizing selection on gene regulation in both species. A deeper understanding of differences in selection on gene expression may be gained by further assessing mean differences, variability, and eQTL contributions in various tissue types across primate groups. Such studies may benefit from single-cell techniques, as we find strong contributions of cell-type heterogeneity in our analysis of variability, which may be biological or technical in nature.

# Materials and methods

## Novel data generation

In total, 39 post-mortem heart tissue biopsies were collected from captive born chimpanzees, 18 of which have been previously described (*Pavlovic et al., 2018*). A partial pedigree suggests at least eight first degree relationships among these individuals. Other metadata, including sex, age at death, and primate research center source of tissue, are detailed in *Figure 1—source data 1*. DNA and RNA were extracted from frozen tissues using kits (QIAGEN Cat No. 74104) or Trizol extraction. RNA-seq and whole genome sequencing libraries were prepared according to manufacturer's protocols (PolyACapture followed by TruSeq v2 RNA Library prep kit; same RNA-seq protocol used by GTEx consortium. Nextera DNA Flex Library prep kit). Sequencing of RNA-seq libraries was performed by University of Chicago sequencing facilities on HiSeq 4000 using 75 bp single-end sequencing chemistry. The 10 RNA-seq libraries previously described (*Pavlovic et al., 2018*) were re-sequenced for additional depth, along with the 29 new libraries. Whole genome sequencing for all 39 chimpanzee samples was performed on NovaSeq using 300+300 paired end sequencing chemistry.

## RNA-seq and differential expression power analysis

39 RNA-seq fastq files for human left ventricle were chosen at random from GTEx v7 (*Figure 1— source data 1*, see Acknowledgements). Additionally, the 10 human and 18 chimpanzee RNA-seq libraries previously generated (*Pavlovic et al., 2018*), and all the novel chimpanzee RNA-seq libraries generated in this study are described in *Figure 1—source data 1*. Fastq files for samples that were sequenced on multiple lanes were combined after confirmation that all gene expression profiles for all fastq files cluster primarily by sample and not by sequencing lane. GTEx-derived fastq files were trimmed to 75 bp single-end reads to match the non-GTEx sequencing data. Reads were aligned to the appropriate annotated genome (GRCh38.p13 or Pan_tro_3.0 from Ensembl release 95; *Zerbino et al., 2018*) using STAR aligner (*Dobin et al., 2013*) default parameters. Only chromosomal contigs were considered for read alignment throughout this work. That is, unplaced contigs were excluded from the reference genome. Gene counts were obtained with subread featureCounts (*Liao et al., 2014*) using a previously described annotation file of human-chimpanzee orthologous exons (*Pavlovic et al., 2018*). The gene expression matrix was converted to CountsPerMillion (CPM) with edgeR (*Robinson et al., 2010*) to normalize reads to library size. The mean-variance trend was estimated using limma-voom (*Ritchie et al., 2015*). Genes with less than 6 CPM in all samples were excluded from further analysis. This cutoff was chosen based on visual inspection of the voom mean-variance trend to identify where the trend becomes unstable. To normalize differences in orthologous exonic gene size between human and chimpanzee, we converted the log(CPM) matrix to log (RPKM) based on the species-specific length of orthologous exonic regions. We then visually inspected PCA plots and hierarchical clustering to identify potential outliers and batch effects (*Figure 1*). We note that although all the chimpanzee RNA isolation and sequencing were prepared separately from GTEx heart samples, PCA and clustering analysis suggest that the inter-species differences vastly outweigh the technical batch effects (based on the inter-species but within-batch samples sourced from *Pavlovic et al., 2018*). The dataset of 49 human samples and 39 chimpanzee samples was culled to 39 human and 39 chimpanzee samples based on exclusion of 5 obvious

human outliers which did not cluster with the rest and had among the lowest read depths (*Figure 1A–B*). The remaining human samples to exclude to reach a balanced set of 39 human and 39 chimpanzee samples were chosen by excluding the remaining GTEx samples with the lowest mapped read depths. Differential expression was tested using limma (*Ritchie et al., 2015*), using the eBayes function with default parameters and applying Benjamini-Hochberg FDR estimation. For *Figure 1—figure supplement 2*, this process was repeated at varying sample size (sampling with replacement) and at various read depths. Sequencing depth subsampling analysis was performed at the level of bam files to obtain matched numbers of mapped reads across samples and differential expression analyses was repeated.

## Contribution of inter-relatedness chimpanzees to differential expression analysis

A centered genetic relatedness matrix of kinship coefficients (the same as used in chimpanzee cis-eQTL mapping) was used to cluster the 39 chimpanzee samples into groups which have varying levels of inter-relatedness. Clusters were determined using 'hclust' and 'cutree' functions in R with k = 7 clusters and defaults for other parameters. This resulted in the seven clusters depicted in *Figure 1—figure supplement 3*, which were further culled into three clusters of comparable size (each of size n = 4) with relatively high inter-relatedness, and one cluster (n = 13) with relatively low inter-relatedness. Specifically, sample 529 in the purple cluster was dropped as it had the lowest mean intra-cluster pairwise kinship coefficient. Conversely, six samples were dropped from the red cluster based on the presence of a high intra-cluster pairwise kinship coefficient which indicate first degree relatives. The VarianceParition R package (*Hoffman and Schadt, 2016*) was used implement a linear mixed model to assess the contribution of the cluster annotations (which we interpret as a proxy for inter-relatedness), RNA extraction batch, and sex (each as random effects) to explaining logRPKM gene expression for each gene. We also used the cluster groupings to empirically evaluate the degree to which highly inter-related subsamples contribute to DE power: More specifically, the same power analysis procedure described above was used to empirically assess the number of DE genes and false positive rate using four human samples (samples 63145, 62606, 59167, 59263, which were all derived from the same technical batch), and four chimpanzee samples, each drawn without replacement from within a cluster of varying degrees of inter-relatedness. The full distribution of DE results from all 715 possible combinations of four chimpanzees drawn from the lowly related n = 13 cluster was used as a baseline.

## Expression variability estimation

To estimate the mean and variance of gene expression, we assume

$$
\begin{aligned}
x_{ij} \mid x_{i+}, \lambda_{ij} &\sim \mathrm{Poisson}\left(x_{i+} l_j \lambda_{ij}\right) \\
\lambda_{ij} &\sim \mathrm{Gamma}\left(\phi_j, \phi_j/\mu_j\right)
\end{aligned}
$$

where $x_{ij}$ is the number of reads mapping to gene $j$ in sample $i$ ($i = 1, \ldots, n; j = 1, \ldots, p$), $x_{i+} = \sum_j x_{ij}$ is the total number of reads observed in sample $i$, $l_j$ is the effective length (*Pachter, 2011*) of gene $j$, and $\lambda_{ij}$ is the true relative gene expression of gene $j$ in sample $i$. The effective length for each gene, $l_j$, was calculated separately for chimpanzee and human as the length of orthologous exonic regions from which aligned reads were summed to create a count matrix. Under this model, true gene expression values for gene $j$, $\lambda_{1j}, \ldots, \lambda_{nj}$, follow a Gamma distribution with mean $\mu_j$ and variance $\mu_j^2/\phi_j$, implying that the observed counts $x_{1j}, \ldots, x_{nj}$ follow a Negative Binomial distribution with mean $x_{i+} l_j \mu_j$ and overdispersion $1/\phi_j$. This model corresponds to a generalized linear model (*Hilbe, 2014*), which we fit by maximizing the likelihood using the 'glm.nb' function in the R package *MASS*.

We fit a LOESS trend using the 'loess.fit' function in R (with degree = 1) to the mean-overdispersion trend across all genes and considered the residual from the trend as the gene's mean-corrected dispersion. We estimated standard errors for each dispersion estimate by bootstrapping with replacement 1000 times. We estimated bootstrap p-values to test the alternative hypothesis that the absolute difference in dispersion between chimpanzee and human is greater than zero, bootstrapping with replacement 10,000 times. More specifically, we estimated the distribution of the absolute difference in dispersion under the null by performing 10,000 iterations of resamples (n = 39

individuals) from a joint count matrix containing all human and chimpanzee individuals. p-Values for each gene are then defined as the fraction of resamples with an absolute difference greater than what is observed. False discovery rates were estimated using Storey's q-value (*Storey and Tibshirani, 2003*).

## Dispersion and cell-type heterogeneity

Single-cell RNA-seq data were downloaded from the *Tabula Muris* mouse single cell atlas (*Tabula Muris Consortium et al., 2018*). This dataset contains both FACS-based and droplet-based single-cell RNA-seq datasets for adult mouse heart. Only the FACS based heart data were used, as the droplet based data are much sparser by comparison (*Tabula Muris Consortium et al., 2018*). Data were analyzed with Seurat (*Butler et al., 2018*) using the published cell type labels (*Tabula Muris Consortium et al., 2018*). After subsetting cells that contain at least 1000 genes with nonzero counts, the scTransform function was used to obtain a normalized count matrix used for plotting *Figure 4A–B*. A cell-type-specificity score, τ (*Kryuchkova-Mostacci and Robinson-Rechavi, 2016*), was calculated for each gene (only considering one-to-one mouse/human orthologs) by utilizing the nine cell type labels assigned by *Tabula Muris* to sum raw read counts from each cell type to create a pseudo-bulk count matrix. The pseudo-bulk count matrix was converted to CountsPerMillion and subsequently used to calculate τ with open source software (doi:10.5281/zenodo.3558708).

BayesPrism (*Chu and Danko, 2020*) was used to estimate cell types in the bulk RNA-seq datasets used for the RNA-seq dispersion and power analyses. The intersection of genes that are one-to-one mouse-human orthologs and used in DE and bulk dispersion estimation were used to filter the cell-type-labeled mouse scRNA-seq reference gene expression matrix for deconvolution ('run.Ted' function with default parameters) of the 39 chimpanzee and 39 human bulk samples used in DE analysis. This process yielded per-individual cellular proportion estimates and expression estimates for each cell type. The expression estimates were converted to log(CPM) based on the library size of the bulk count matrix. A cell-type-specific dispersion estimate was obtained similarly to the bulk procedure: a LOESS trend was fit to the population mean expression versus the log(variance) trend across all genes, and the residual was considered as the cell-type-specific dispersion estimate. Standard errors for cell-type-specific dispersion estimates were obtained by bootstrapping 1000 samples from the BayesPrism estimated cell-type-specific expression matrices, and as such, the reported standard error (*Figure 4—figure supplement 1—source data 1*) does not incorporate error in cell type deconvolution or expression estimation.

## Chimpanzee genome sequencing

For whole genome sequencing data processing, we followed general guidelines for read alignment and de novo variant calling as previously described (*Li, 2014*). More precise steps are as follows: Sequencing adapters were trimmed with cutadapt (*Martin, 2011*) and aligned to Pan_tro_3.0 (Ensembl) using bwa-aligner (*Li and Durbin, 2009*) with default parameters. The average genome coverage in every sample was >30X. Sample-specific statistics for basic data processing steps, including read alignment and variant calling, are summarized in *Supplementary file 2*. PCR duplicates were removed via Picard tools. Low-complexity regions of the genome were determined using dustmasker (*Morgulis et al., 2006*) with default settings and excluded from variant calling. Sites where any sample had coverage (after PCR duplicate removal) outside of $d+3\sqrt{d}$ and $d+4\sqrt{d}$ coverage (where d is the sample average fold coverage across the genome) were also excluded, as these regions are enriched for duplicated or paralogous regions which are prone to false heterozygous variant calls (*Li, 2014*). The resulting callable sites span 2,544,417,587 out of 2,967,125,077 bases on the contiguous chromosomal genome. Variants were called in all samples jointly using freebayes (*Garrison and Marth, 2012*) with the following parameters: {`-min-coverage` 3 `-max-coverage` 150 k –standard-filters -n 2 –report-genotype-likelihood-max} and subsequently filtered for Phred-scaled quality score >30 to generate a VCF file (see Data Availability). Due to memory constraints, variant calling was executed in 2.5 megabase chunks which were later merged. In total, 19,789,407 single nucleotide variants passed variant calling filters, yielding a transition/transversion ratio of 2.08. Variants in the accompanying VCF and throughout this work were left 'clumped', meaning that completely linked SNPs within 3 bp (the default setting of freebayes) are combined into a single variant in the VCF, and tested as a single variant during eQTL calling.

For admixture analysis, a VCF file of previously sequenced wild-born chimpanzees (*de Manuel et al., 2016*) was lifted over to Pan_tro_3.0 and merged with the VCF file described above, keeping only variants present in both sets. Variants in high LD were pruned using plink (*Purcell et al., 2007*) with parameters {–indep-pairwise 50 5 0.5}. The resulting genotypes were analyzed for population structure with PCA using 'prcomp' function in R. Additionally, we utilized Admixture software (*Alexander et al., 2009*) with K = 4 clusters, as there are four recognized distinct chimpanzee sub-species, all of which were represented in the wild-born cohort (*de Manuel et al., 2016*).

## Cis eQTLmapping (chimpanzee)

RNA-seq reads were re-aligned with STAR aligner (*Dobin et al., 2013*) using Pan_tro_3.0 gene annotations from Ensembl version 95 (*Zerbino et al., 2018*). Gene counts were quantified with STAR – quantMode GeneCounts to compile a gene expression matrix. Genes were filtered to require at least eight read counts in 75% of samples, leaving 13,545 genes for further analysis. The resulting read count matrix was converted to log(CPM), standardized across individuals, and quantile normalized to a normal distribution across genes as previously described (*Degner et al., 2012*). Principle component analysis of the normalized matrix identified significant associations between various observed technical factors and some of the first 10 principle components, including RNA library prep batch and sex, and as such, principle components were included later as covariates during eQTL calling.

Variants were filtered for MAF >0.1 and Hardy-Weinberg equilibrium (nominal $P>1\times10^{-7.5}$, hardy function in plink) to filter out rare variants and genotyping errors. The resulting 5,957,179 variants were each tested for association with expression of each local *cis* gene (*cis* window defined as within 250 kb of gene body). MatrixEQTL (*Shabalin, 2012*) was used to implement a linear mixed model for each *cis* variant:gene pair to estimate the effect of the variant genotype on normalized expression gene expression. We supplied MatrixEQTL with a genetic relatedness matrix (GRM) to account for heteroskedastic errors generated from underlying population structure and genetic relatedness amongst individuals. The standardized GRM was produced by GEMMA (*Zhou and Stephens, 2012*) using variants pruned for LD as described for admixture analysis. Additionally, between 0 and 15 gene expression principle components (PCs) were tested as covariates to the linear model and 10 PCs were included in the final model as this maximizes the number of eQTLs (*Figure 6—figure supplement 1A*). Manual inspection of normalized expression boxplots stratified by genotype for the top eQTLs revealed many of the nominally strongest associations were driven by a single expression outlier point for a single homozygous individual, MD_And. Further inspection revealed this individual has among the highest levels of homozygosity genome-wide among our chimpanzee cohort (*Supplementary file 2*), possibly due to inbreeding. Given that this individual is the only chimpanzee sourced from MD Anderson primate research center, we felt justified excluding this individual from eQTL calling to minimize false associations. After excluding this individual and re-performing eQTL mapping (n = 38), visual inspection of a QQ-plot of p-values compared to permuted null data (where the sample labels for expression and covariates were randomly assigned to genotype) indicates inflation of small p-values, and that p-values are well calibrated under the permuted null (*Figure 6—figure supplement 1B*). To obtain gene-level p-values that test whether a gene contains an eQTL (eGenes) we used EigenMT, a method which approximates permutation testing procedures to account for multiple testing of linked SNPs (*Davis et al., 2016*).

## Cis eQTL mapping (human)

We downloaded gene-level (eGene) summary statistics for Heart_Left_ventricle GTEx v8 from GTEx portal (https://gtexportal.org/). The summary statistics from this mapping pipeline only considers expressed genes, which are defined as >0.1 TPM in at least 20% of samples and ≥6 reads in at least 20% of samples. The analysis in *Figure 6—figure supplement 3B* required more than summary statistics. We downloaded normalized phenotypes, covariates, and genotypes for GTEx v8 data for Heart_Left_ventricle and remapped eGenes using varying sample sizes using a mapping pipeline nearly identical to GTEx. Namely, we used FastQTL (*Ongen et al., 2016*) on the supplied data with randomly selected individuals corresponding to sample sizes (n) of 40, 60, 80, 100, 120, 160, 200. Similar to guidelines described by GTEx (https://gtexportal.org/), we included only 10 PEER factor covariates for n = 40, 15 for 40 > n > 150, or 30 for n > 150.

## Gene-wise conservation statistics

Gene-wise amino acid percent identity between chimpanzee and human was obtained from BioMart (*Kinsella et al., 2011*). Gene-wise pan-mammal dN/dS for each gene was obtained from a previous study that used alignments from 29 mammals (*Lindblad-Toh et al., 2011*). $P_n/P_s$ was calculated from all GTEx v8 genotype data and the union of all *Pan troglodyte* genotypes available in this study and *de Manuel et al., 2016*. Specifically, Ensembl vep (*McLaren et al., 2016*) was used to annotate coding variants with MAF >0.1 as synonymous or non-synonymous to tabulate Pn (number of polymorphic non-synonymous sites) and Ps (number of polymorphic synonymous sites) for each gene within each species. After requiring that genes have at least one polymorphism in both species, a pseudo-count of 0.5 was added to both Pn and Ps for each gene for both species to avoid division-by-zero errors. TATA box genes were classified as genes with a TATA motif within 35 bp of a transcription initiation site from published transcription initiation sites (*Abugessaisa et al., 2019*).

## eQTL-mapping of shared polymorphisms

263 SNPs among 125 regions that are trans-species polymorphisms between chimpanzee and humans were obtained from *Leffler et al., 2013*. Each trans-species polymorphic *region* contains at least two trans-species SNPs to ensure regions are identical by descent rather than recurrent mutation of an isolated SNP (*Leffler et al., 2013*). Only SNPs with MAF >0.1 in our datasets were further utilized, leaving 192 SNPs in chimpanzee, 196 SNPs in human (GTEx), and 144 SNPs (amongst 76 regions) tested for eQTL activity in both species. For each test SNP, a matched control SNP was randomly chosen for each species with the criteria that it should have a matching allele frequency (±5% MAF), within 100 kb of the test SNP, and unlinked to the test SNP ($R^2$ <0.2, LD calculated with plink). We used MatrixEQTL to retest these SNPs for *cis* eQTL activity (1 MB window) using MatrixEQTL with the same normalized gene expression matrix, GRM matrix (for chimpanzee only), and covariates described above for chimpanzee and human *cis* eQTL mapping. The effect sizes between species occasionally had to be re-polarized to relate to the same allele, as the effect sizes obtained from eQTL testing software is often polarized by minor allele or by reference vs non-reference allele, although the minor allele and/or reference allele at these trans-species polymorphisms is not always the same between species.

## Gene set enrichment analysis

Gene set enrichment and gene ontology overlap analysis was performed with clusterProfiler R package (*Yu et al., 2012*). The GSEA test was performed with an ordered gene list using the 'gseGO' function with 1,000,000 permutations. As ordering genes was not applicable to inter-species eGene classifications, the 'enrichGO' function was used to perform gene ontology overlap analysis (hypergeometric test) with foreground and background gene sets based on eGene classifications. To quantify the correlation between dN/dS and dispersion for different GO sets, we obtained dN/dS annotations as described above (*Lindblad-Toh et al., 2011*) and gene set annotations from MSigDB v7.2 (*Liberzon et al., 2011*). We tested all gene sets with dN/dS and dispersion estimates for at least five genes in the gene set using Spearman's test, adjusting for multiple testing with the Storey's Q-value (*Storey and Tibshirani, 2003*).

## Acknowledgements

We thank Natalia Gonzales, Michelle Ward, Ittai Eres, and other members of the Gilad lab for helpful analysis discussions and comments on the manuscript. This work was supported by NIH grant R35GM131726 as well as the Yerkes National Primate Research Center Base Grant ORIP/OD P51OD011132 and RR00165,. Computational resources were provided by the University of Chicago Research Computing Center.

The Genotype-Tissue Expression (GTEx) Project was supported by the Common Fund of the Office of the Director of the National Institutes of Health, and by NCI, NHGRI, NHLBI, NIDA, NIMH, and NINDS. The data used for the analyses described in this manuscript were obtained from: the GTEx Portal on 10/08/19 and/or dbGaP accession number phs000424.v7.p2 on 09/17/19.

## Additional information

### Funding

| Funder | Grant reference number | Author |
|---|---|---|
| National Institute of General Medical Sciences | R35GM131726 | Yoav Gilad |

The funders had no role in study design, data collection and interpretation, or the decision to submit the work for publication.

### Author contributions
Benjamin Jung Fair, Conceptualization, Data curation, Formal analysis, Visualization, Methodology, Writing - original draft, Writing - review and editing; Lauren E Blake, Resources, Data curation, Writing - review and editing, Discussion and interpretation of results; Abhishek Sarkar, Methodology, Writing - review and editing, Discussion and interpretation of results; Bryan J Pavlovic, Resources, Data curation, Investigation; Claudia Cuevas, Investigation; Yoav Gilad, Conceptualization, Supervision, Funding acquisition, Writing - original draft, Writing - review and editing

### Author ORCIDs
Benjamin Jung Fair (iD) https://orcid.org/0000-0001-6296-5703
Bryan J Pavlovic (iD) http://orcid.org/0000-0002-7751-5315
Yoav Gilad (iD) https://orcid.org/0000-0001-8284-8926

### Decision letter and Author response
Decision letter https://doi.org/10.7554/eLife.59929.sa1
Author response https://doi.org/10.7554/eLife.59929.sa2

## Additional files

### Supplementary files
• Supplementary file 1. Full GSEA results based on interspecies dispersion differences after excluding virally challenged chimpanzees.

• Supplementary file 2. Whole genome sequencing sample summary statistics.

• Supplementary file 3. Chimpanzee eGene summary statistics.

• Transparent reporting form

### Data availability
RNA-Seq data available under GEO accession number GSE151397. Raw whole genome sequencing data under SRA accession PRJNA635393. Processed whole genome sequencing data available as variant calls at European variation archive, EVA accession PRJEB39475.

The following datasets were generated:

| Author(s) | Year | Dataset title | Dataset URL | Database and Identifier |
|---|---|---|---|---|
| Fair BJ, Blake LE, Chavarria C, Sarkar A, Pavlovic BJ, Gilad YY | 2020 | Gene expression variability in human and chimpanzee populations share common determinants | https://www.ncbi.nlm.nih.gov/geo/query/acc.cgi?acc=GSE151397 | NCBI Gene Expression Omnibus, GSE151397 |
| Fair BJ, Blake LE, Chavarria C, Sarkar A, Pavlovic BJ, Gilad YY | 2020 | Whole genome sequencing of 39 captive born chimpanzees | https://www.ncbi.nlm.nih.gov/bioproject/PRJNA635393/ | NCBI BioProject, PRJNA635393 |
| Fair BJ | 2020 | Whole genome sequencing of 39 captive born chimpanzees | https://www.ebi.ac.uk/eva/?eva-study=PRJEB39475 | EBI European Variation Archive, PRJEB39475 |

The following previously published datasets were used:

| Author(s) | Year | Dataset title | Dataset URL | Database and Identifier |
|---|---|---|---|---|
| Pavlovic BJ, Blake LE, Chavarria C, Gilad Y | 2018 | A Comparative Assessment of iPSC Derived Cardiomyocytes with Heart Tissues in Humans and Chimpanzees | https://www.ncbi.nlm.nih.gov/geo/query/acc.cgi?acc=GSE110471 | NCBI Gene Expression Omnibus, GSE110471 |
| The GTEx Consortium | 2019 | GTEx Analysis V8 | https://www.gtexportal.org/home/datasets | dbGaP, phs000424.v8.p2 |

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
