## [Decision Letter]

**Acceptance summary:**

Individuals show heritable variation in gene expression, which is associated with disease susceptibility. In contrast to many other studies that have focused on mapping associations between genetic and gene regulatory variation, the current work addresses group dispersion/variance of gene expression among samples as well as the evolutionary processes that shape differences in gene expression between individuals, in both humans and chimpanzees. Using computational deconvolution, the authors demonstrate that cell-type heterogeneity is an important component of expression variability, with significant overlap of orthologous genes associated with eQTLs in both species. The conclusion is that gene expression variability in humans and chimpanzees often evolves under similar evolutionary pressures.

**Decision letter after peer review:**

Thank you for submitting your article "Gene expression variability in human and chimpanzee populations share common determinants" for consideration by *eLife*. Your article has been reviewed by three peer reviewers, and the evaluation has been overseen by a Reviewing Editor and Detlef Weigel as the Senior Editor. The following individual involved in review of your submission has agreed to reveal their identity: Charles G Danko (Reviewer #3).

The reviewers have discussed the reviews with one another and the Reviewing Editor has drafted this decision to help you prepare a revised submission.

Summary:

This is a solid study, with a large sample size, identifying quantitative trait loci (eQTLs) in humans and chimpanzees, using gene expression data from primary heart samples. The authors complemented the analysis of gene expression with a comparative eQTL mapping, as opposed to relying on mean expression levels, as most comparative studies like this one do. Also unlike many studies focused on mapping associations between genetic and gene regulatory variation, the authors paid attention to the group dispersion/variance of gene expression among samples as well as the evolutionary processes that shape the differences in gene regulation between individuals. The calculation of power for discovering differentially expressed genes as a function of sample size at the beginning of the paper is a thoughtful analysis that is useful to many in the community. All of the analyses are extremely thorough and well-executed. The statistical tests are appropriate and rigorous. Results are interpreted in a conservative fashion.

The main limitation is that the authors are not able to conclusively disambiguate between different causes of dispersion. Genetics, cell type, and technical variation may all contribute to dispersion. The authors state this very clearly throughout the manuscript. In part, this may reflect the authors' underselling their results somewhat. But in part, this really does reflect reality: Cell type is a major confounder that may provide false signals in other analyses.

The reviewers suggested a number of potential additions to clarify current results or build upon them. I will leave it up to the authors to decide which are worth including in their revision.

1) The first test authors conducted is to identify differentially variable (DV) genes. A total of 2658 DV genes were identified. The problem of the result is that almost equal number of up- and down-regulated DV genes symmetrically distributed around DV=0. Often, this is an indication of a lack of biological signals in data analysis. This might be due to the pooling of gene groups with diverse functionality together. Therefore, this reviewer suggests that authors should break down genes into subgroups to detail the up and down-regulatory patterns with the hope that some of the gene groups give interpretable results

2) The second test is to correlate the higher coding sequence conservation with lower dispersion. Again, the positive result is not unexpected. There are many indirect and/or confounding factors that may explain the effect. This reviewer, however, understands it is impossible to control them all (also authors have attempted to address some of them in the next few tests). However, here it is better to add exploratory analyses for genes in different functional groups and also give examples of outlier genes that do not follow the rule.

3) The third test is to examine the correlation between gene expression variability with single-cell type heterogeneity of samples. Authors first used Tabula Muris dataset to show dispersion is strongly correlated with cell-type specificity/diversity. If this is true, then the point that authors really wanted to demonstrate is, in fact, hampered. Authors might really want to show the "true" single-cell variability (see, for example, PMID: 31861624) is correlated with the level of group variance of gene expression.

4) The fourth test authors conducted is to show that dN/dS and P_n_/P_s_ ratios of genes are correlated with gene expression variability (variance). However, because of the existence of heterogeneity of cell-type composition in samples, any correlation observed may be utterly biased by this single uncontrollable confounding factor. Furthermore, heart tissues contain an over-abundant expression of genes encoded in the mitochondrial genome. The expression level of these mt-genes may vary substantially between samples and reflect the health status of primary sample donors. PEER normalization may have to take this into account as a covariant.

5) Several other tests authors performed are around eQTLs (eGene overlap and eSNP overlap) between the two species. These are typical tests evolutionary biologists usually try to do whenever data are available. However, the issues with these types of tests are the low power in general. More importantly, in order to be consistent with previous tests which are all around the explanation of gene expression variance, this part should address the overlap between expression vQTLs in humans and chimps.

6) I would like to see more discussion about the inter-relatedness of the chimpanzees in the analysis of gene expression. Is that contributing to the power of the DE analysis, which has really high numbers of DE genes. That may certainly be due to the large samples size, but should be addressed. Related to that, the support that the gene-wise dispersion estimates are well correlated in humans and chimpanzees overall (Figure 1C, and Figure 2—figure supplement 1) seems qualitative. It looks like the chimpanzees might have less dispersion overall?

7) What do the authors think these findings mean for study systems outside of humans and captive chimpanzees? Both on the technical level (e.g. sample size), and for how their approach could be helpful outside of these species. Generalizing this approach would broaden the impact and audience of the paper.

8) Did the authors test directly whether eQTLs were enriched in genes with a high dispersion? I could not find this going back through the paper. This seems almost trivially likely to be true. I may have missed this result? Or did the authors worry this is too likely to be confounded with cell type? Either way, this seems like a result that may be useful to show even if the authors did acknowledge that it was likely to be confounded.

9) Did the authors consider looking for cell-type QTLs? They state several times in the paper the possibility that genetic factors may influence cell types. They have enough data – at least in human – to obtain QTLs for specific cell types, as others have done (Marderstein et al., 2020; Donovan et al., 2020). If these cell type QTLs were enriched near genes with a high dispersion, this may bolster the author's argument that genetic factors underlie dispersion by affecting cell type composition.

10) The scRNA-seq reference used for estimating cell types in heart tissue was derived from mice. Could this lead the authors to underestimate the degree to which cell types drive dispersion in genes that are variable between human and chimp? Genes that are variable between human/ chimp may also be more likely to be variable between either species and mouse, and perhaps this variability has led to them becoming more/ less of a marker of a specific cell population (and hence their dispersion in primates does not correlate with cell type specificity in mouse).

11) Have the authors tried estimating dispersion on top of what is expected based on differences in cell type? There are several strategies that might work for this: There are new strategies for estimating a posterior of cell type specific expression from a bulk sample, conditional on scRNA-seq data as prior information (Chu and Danko, 2020). These cell type specific expression estimates could then be analyzed for dispersion. Alternatively, it may also work to regress the estimated proportion of each cell type out of the dispersion estimates. While there are certainly a lot of pitfalls with using these strategies, especially in the setting shown here (all of this would work better if there were species matched reference data), they might provide an avenue for depleting the contribution of cell type differences from dispersion estimates.

12) Can the authors add a dotted line to show the shape of the distribution for genes with low dispersion, or where dispersion is shared in both human and chimpanzee, in Figure 4B? Is this different from genes that are dispersed in either chimp or human?

---

## [Author Response]

Revisions for this paper:The reviewers suggested a number of potential additions to clarify current results or build upon them. I will leave it up to the authors to decide which are worth including in their revision.1) The first test authors conducted is to identify differentially variable (DV) genes. A total of 2658 DV genes were identified. The problem of the result is that almost equal number of up- and down-regulated DV genes symmetrically distributed around DV=0. Often, this is an indication of a lack of biological signals in data analysis. This might be due to the pooling of gene groups with diverse functionality together. Therefore, this reviewer suggests that authors should break down genes into subgroups to detail the up and down-regulatory patterns with the hope that some of the gene groups give interpretable results

We thank the reviewers for their helpful comments, which ultimately improved the manuscript.

Regarding this point, it is true that symmetric destitutions are expected when there is a lack of biological signal, but we also often see such distributions when there is abundant biological signal, especially when we compare quantitative traits between species. For example, it is well established that among inter-species differentially expressed genes in practically any comparison that was done to date, about half the genes have elevated expression levels and about half have decreased expression.

That said, we provide the analysis suggested by the reviewer (revised manuscript Figure 4—figure supplement 3). We performed GSEA enrichment to identify which gene categories are preferentially DV-up or DV-down between chimpanzee and human. We found immune related genes with higher dispersion in chimpanzee, and mitosis related genes with higher dispersion in humans. As described in the manuscript, our interpretation is that this partly reflects differences in life histories in the human population, from which many individuals have suffered cardiac ischemia and may have differentially expressed mitosis related genes as a response.

2) The second test is to correlate the higher coding sequence conservation with lower dispersion. Again, the positive result is not unexpected. There are many indirect and/or confounding factors that may explain the effect. This reviewer, however, understands it is impossible to control them all (also authors have attempted to address some of them in the next few tests). However, here it is better to add exploratory analyses for genes in different functional groups and also give examples of outlier genes that do not follow the rule.

This is a good suggestion. In the revised version, we provide a new analysis (revised manuscript Figure 3—figure supplement 1) in which we test for a correlation between higher coding sequence conservation (dN/dS metric) with lower dispersion on a GO category basis. Consistent with the narrative throughout this manuscript, we find that this correlation is strongest for immune related genes.

3) The third test is to examine the correlation between gene expression variability with single-cell type heterogeneity of samples. Authors first used Tabula Muris dataset to show dispersion is strongly correlated with cell-type specificity/diversity. If this is true, then the point that authors really wanted to demonstrate is, in fact, hampered. Authors might really want to show the "true" single-cell variability (see, for example, PMID: 31861624) is correlated with the level of group variance of gene expression.

This is an important question that we have pondered ourselves as we wrote this manuscript. This comment is similar to reviewer point 11, and we provide description of a new analysis to address this point in our response to point 11 (please read below).

4) The fourth test authors conducted is to show that dN/dS and P_n_/P_s_ ratios of genes are correlated with gene expression variability (variance). However, because of the existence of heterogeneity of cell-type composition in samples, any correlation observed may be utterly biased by this single uncontrollable confounding factor. Furthermore, heart tissues contain an over-abundant expression of genes encoded in the mitochondrial genome. The expression level of these mt-genes may vary substantially between samples and reflect the health status of primary sample donors. PEER normalization may have to take this into account as a covariant.

We do not understand the first concern – that dN/dS and P_n_/P_s_ may be confounded by the presence of cell type composition heterogeneity. Obviously, dN/dS and P_n_/P_s_ are based on genotypes, and as such are orthogonal to gene expression based measurements from tissues (because all cell types from the same individual have the same genotypes). Given this, how can these measurements be biased by cell composition?

The second point – that mt-genes may be important markers of cellular or organismal fitness that might be important to correct for – is an interesting possibility and we further investigated it. Here, (Author response image1) we show that the first 10 gene expression principal components, which we include as covariates in the eQTL linear models, explain a nearly identical fraction of the total variance whether or not we include mt-genes in the gene expression matrix (the exact comparison is PCs from a gene expression matrix in which MT-genes are included, versus one where the median expression of MT-genes is set to the median across all samples, simulating zero variance contributed by MT-genes). Therefore, we believe our original principal component covariates are sufficient to capture the mitochondrial gene expression expression components that may reflect health status, and we did not alter the eQTL model or results from those in our original submission.

**Author response image 1. sa2fig1:** 

5) Several other tests authors performed are around eQTLs (eGene overlap and eSNP overlap) between the two species. These are typical tests evolutionary biologists usually try to do whenever data are available. However, the issues with these types of tests are the low power in general. More importantly, in order to be consistent with previous tests which are all around the explanation of gene expression variance, this part should address the overlap between expression vQTLs in humans and chimps.

We agree that eGene overlap analyses are limited by power, as eQTL analyses often require much larger sample sizes to detect modest eQTL effects. While we are fundamentally limited by the data available to us, we do address the overlap between dispersion and eGenes (revised Figure 6B, Figure 6—figure supplement 4). Unfortunately addressing any potential inter-species overlap with varianceQTLs (vQTLs) would require sample sizes larger than we can obtain. For example, [Sarkar et al., PLoS Genet. 2019] suggests a sample sizes in the thousands required to identify genetic variants that explain variance independent of mean expression.

6) I would like to see more discussion about the inter-relatedness of the chimpanzees in the analysis of gene expression. Is that contributing to the power of the DE analysis, which has really high numbers of DE genes. That may certainly be due to the large samples size, but should be addressed. Related to that, the support that the gene-wise dispersion estimates are well correlated in humans and chimpanzees overall (Figure 1C, and Figure 2—figure supplement 1) seems qualitative. It looks like the chimpanzees might have less dispersion overall?

Thank you for the good suggestion regarding the contribution of inter-related chimpanzees to DE analysis. We performed a set of new analyses to address this (revised manuscript Figure 1—figure supplement 3). First, we identified clusters of our chimpanzee samples based on genetic relatedness, identifying groups of inter-related chimpanzees and unrelated chimpanzees as test and control groups to investigate further. We note that qualitatively, the genomewide expression measurements of chimpanzee individuals that are more closely related are not more similar than that of unrelated individuals, likely due to confounding factors in the data, such as the batch effect of RNA isolation or other technical effects. We further used the VariancePartition R package to quantify the effects of inter-relatedness and technical batch effects, finding that technical batch effects likely have a greater effect on gene expression and DE analysis. Finally, we empirically address this question by reperforming DE analysis with the test (inter-related chimpanzee samples) and control (unrelated samples) and quantifying the number of DE genes and estimated false positive rate. We do not find any meaningful differences that would point to inter-related samples having effects of the similar magnitude as the technical batch effects.

7) What do the authors think these findings mean for study systems outside of humans and captive chimpanzees? Both on the technical level (e.g. sample size), and for how their approach could be helpful outside of these species. Generalizing this approach would broaden the impact and audience of the paper.

Referring to our analysis of DE power, this is an excellent question, but empirically answering this question is outside the scope of this paper so we can only speculate. Though, we reason that our findings may depend on the species comparison (species which have diverged more may have greater differences, and thus a similarly sized study may identify more differences), and the tissue type (tissues with high inter-individual variability due to technical or environmental factors may have less power). We have added a discussion of this in the revised manuscript. Further, we provide an analysis of GTEx tissues which quantifies the variability (gene-wise overdispersion parameter estimate measured from a negative binomial) across GTEx samples for different tissues (Author response image 2, top). We note that the median amount of population overdispersion within a tissue negatively correlates with the number of eGenes detected in GTEx (independent of GTEx sample size, Author response image 2 bottom left, bottom right). We believe this may serve as a reference as to which tissues may have more power in DE analysis or eQTL analysis due to the degree of inter-individual variability. We have decided not to include this analysis in the main manuscript as we feel it does not easily fit into the focus of existing narrative on inter-species differences in variability and eQTLs.

8) Did the authors test directly whether eQTLs were enriched in genes with a high dispersion? I could not find this going back through the paper. This seems almost trivially likely to be true. I may have missed this result? Or did the authors worry this is too likely to be confounded with cell type? Either way, this seems like a result that may be useful to show even if the authors did acknowledge that it was likely to be confounded.

Yes. As the reviewer hypothesizes, eQTL containing genes (eGenes) have higher dispersion than non-eGenes. We have added a new figure (revised manuscript Figure 6—figure supplement 4) to show this. Related to this point, in the initial submission we showed that the genes that are eGenes specifically in chimpanzee and not human, have higher dispersion in chimpanzee than human, and vice versa (revised manuscript Figure 6B).

9) Did the authors consider looking for cell-type QTLs? They state several times in the paper the possibility that genetic factors may influence cell types. They have enough data – at least in human – to obtain QTLs for specific cell types, as others have done (Marderstein et al., 2020; Donovan et al., 2020). If these cell type QTLs were enriched near genes with a high dispersion, this may bolster the author's argument that genetic factors underlie dispersion by affecting cell type composition.

Thank you for the good suggestion. Here, we performed the reviewer’s suggested analysis (Author response image3). We used cell composition estimates from [Donovan et al., 2020] on GTEx heart left ventricle samples and performed a genome-wide association study in an attempt to identify variants that associate with cell type composition (similar to [Marderstein et al., 2020]). We used a linear mixed model with a genetic relatedness matrix to account for ancestry, included sex as a covariate, and used the quantile normalized proportion of cardiac muscle cells as the response variable. No loci of achieved the standard GWAS stringent genome wide significance threshold of 5E-8 (Author response image 3, top), but nonetheless, we asked whether the top 100 loci (estimated FDR<0.5) are closest to highly dispersed or lowly dispersed genes (Author response image 3, lower left). Furthermore, under the hypothesis that eQTLs may reflect direct effects on cell type composition, we asked whether eQTLs (summary statistics obtained from GTEx) have inflated P-values for association with cell type composition (Author response image 3, lower right). We did not find any meaningful effect, suggesting that the GTEx mapping pipeline does a good job at accounting for cell type heterogeneity through PEER, and/or lack of power for detecting such cell type QTLs.

**Author response image 3. sa2fig3:** 

Our analysis in response to this point led us to an additional analysis which is not directly related to this reviewer’s point, but nonetheless we have included in the revised manuscript (revised manuscript Figure 4—figure supplement 2) and we will briefly summarize here: The cell type composition estimates from [Donovan et al., 2020] include all GTEx heart samples (from both left ventricle, and atrial appendage tissues, often from the same set of individuals). We used this as an opportunity to examine whether the gene expression variability due to cell type composition may have a strong genetic or individual component, as opposed to being a completely technical artifact of inconsistencies in tissue dissection and sample acquisition. We reasoned that if intentionally anatomically different heart sections (left ventricle, versus atrial appendage) from the same individual correlate better than matched tissue samples from different individuals, then the cell type composition differences across our chimpanzee samples likely are driven by individual level differences, rather than technical differences in sample acquisition. In Figure 4—figure supplement 1-A we show qualitatively that there is an obvious individual level correlation between the atrial appendage samples and left ventricle samples for matched individuals. We used a linear mixed model (implemented with VariancePartition R package) to quantify the contribution of individual level versus tissue level factors to cell type composition estimates, and find that the individual level factor generally explains more variance. We interpret this as strong evidence that the sample to sample differences captured by our dispersion estimates are driven largely by true differences between individuals, rather than random technical differences in sample acquisition and dissection.

10) The scRNA-seq reference used for estimating cell types in heart tissue was derived from mice. Could this lead the authors to underestimate the degree to which cell types drive dispersion in genes that are variable between human and chimp? Genes that are variable between human/ chimp may also be more likely to be variable between either species and mouse, and perhaps this variability has led to them becoming more/ less of a marker of a specific cell population (and hence their dispersion in primates does not correlate with cell type specificity in mouse).

Good point! We agree that there is likely a general downwards bias (ie regression dilution) in the estimated effect of cell type specificity on dispersion in primates due to the cell type specificity estimates being based on mouse single cell data. We added this point in the revised manuscript.

11) Have the authors tried estimating dispersion on top of what is expected based on differences in cell type? There are several strategies that might work for this: There are new strategies for estimating a posterior of cell type specific expression from a bulk sample, conditional on scRNA-seq data as prior information (Chu and Danko, 2020). These cell type specific expression estimates could then be analyzed for dispersion. Alternatively, it may also work to regress the estimated proportion of each cell type out of the dispersion estimates. While there are certainly a lot of pitfalls with using these strategies, especially in the setting shown here (all of this would work better if there were species matched reference data), they might provide an avenue for depleting the contribution of cell type differences from dispersion estimates.

Thanks for helpful suggestion! Though we agree that there are naturally a lot of caveats to such an analysis (estimating cell type specific expression and dispersion from bulk based on a mouse reference), in the revised manuscript, we describe a new analysis (Figure 4—figure supplement 1) using the suggested methodology of [Chu and Danko, 2020]: After estimating cell type composition and expression on a per individual basis, we then estimated dispersion on a per cell type basis in both human and chimpanzee. We present the results as a correlation matrix with hierarchal clustering. We find that while cell type specific expression measurements cluster by cell type before species, cell type specific dispersion estimates cluster in a complex pattern that is more strongly influenced by species. We find this consistent with the idea that dispersion across a population of bulk samples is meaningfully influenced by both cell type composition and genetic effects. In other words, when you are able to correct for the cell type composition effects, or rather, to estimate dispersion within a controlled cell type, the influence of genetic effects is made more obvious. As such, the chimpanzee and human cell specific dispersion estimates cluster partly by species, as the variation due to genetic variants is expected to almost completely segregate by species.

Furthermore, as we felt the cell type decomposition step of this new analysis was unnecessarily redundant with the cell type decomposition estimates we previously presented using CIBERSORT algorithm, we replaced the original figure and methods description referencing CIBERSORT cell decompositions with the cell decomposition estimates from the methodology of [Chu and Danko, 2020] (revised manuscript Figure 4—figure supplement 2). The cell type proportion estimates between the two methods are generally well correlated (R^2^=.88 across all individuals and cell types) and the small discrepancies do not alter any primary conclusions of the paper.

12) Can the authors add a dotted line to show the shape of the distribution for genes with low dispersion, or where dispersion is shared in both human and chimpanzee, in Figure 4B? Is this different from genes that are dispersed in either chimp or human?

We added a scatter plot inset in revised manuscript Figure 4B and Figure 4A to show the distribution dispersion estimates for the plotted genes in both human and chimp.